

# Lability of natural organic matter in freshwater: a simple method for detection using hydrogen peroxide as an indicator

Isabela Carreira Constantino[a*], Amanda Maria Tadini[a], Marcelo Freitas Lima[a], Lídia Maria de Almeida Plicas[a], Altair Benedito Moreira[a], Márcia Cristina Bisinoti[a]

[a]São Paulo State University (UNESP), Instituto de Biociências, Letras e Ciências Exatas, Campus de São José do Rio Preto, Departamento de Química e Ciências Ambientais, R. Cristóvão Colombo 2265, 15054-000 São José do Rio Preto – SP, Brasil.

*Correspondence to:* Isabela C. Constantino (e-mail: isabela.carreira@gmail.com)

## Abstract

Natural organic matter (NOM) is an important component for understanding the behavior of pollutants in the environment. A fraction of NOM is considered labile, fresh and less oxidized. In this work, a simple method was developed to distinguish between labile (LOM) and recalcitrant (ROM) organic matter in freshwater samples. Pyruvate, lignin and fulvic acid were chosen as model compounds of labile and recalcitrant NOM. The samples were submitted to kinetic monitoring experiments using hydrogen peroxide. Pyruvate was the best standard for the quantification of LOM (for concetrations up to 2.9 mg L$^{-1}$). ROM was quantified by measuring the difference between total organic carbon (TOC) and LOM concentrations. Curves obtained with 0.5 to 5.0 mg L$^{-1}$ TOC (pyruvate) in freshwater or ultrapure water samples did not indicate the existence of a matrix effect. This simple method was applied to water samples that were collected monthly for one year; the resulting LOM concentrations ranged from 0.47 to 2.1 mg L$^{-1}$ and the ROM concentrations ranged from 0.08 to 3.5 mg L$^{-1}$. Based on this results we concluded that hydrogen peroxide kinetics can be used as a simple method to quantify LOM and ROM concentrations in freshwater samples.

*Keywords: lability, hydrogen peroxide, organic matter, recalcitrant, freshwater.*

## 1 Introduction

Natural organic matter (NOM) is a key component in aquatic environments. It originates from allochthonous formations in soil and by chemical or microbiological decay of animal and vegetal tissues. It can also stem from autochthonous production in water (Lindell; Granéli; Tranvik, 1995; Vanloon and Duffy, 2005; Uyguner-Demirel and Bekbolet, 2011). NOM in aquatic environments plays important roles in regulating the dynamics of species and in other processes, such as biogeochemical cycles, complexation, photochemical reactions, and carbon





cycling (McKnight, et al. 1985; Melo et al. 2012; Nebbioso and Picollo 2013; Thurman et al. 1982; White; Vaughan; Zeep, 2003; Westhorpe; Mitrovic, 2012).

40       The chemical composition of NOM in aquatic environments is complex and variable, due to different sources of precursor material and the biogeochemical processes involved. The content of non-humic entities is represented by low-molecular weight organic compounds, such as fatty acids, amino acids and aldehydes (Lindell; Granéli; Tranvik, 1995; Paul et al, 2004; Catalán et al, 2013; Xiao et al, 2013). According to some studies, nearly 80% of NOM is

composed of recalcitrant fractions and humic substances (HS) (Leenheer; Croué, 2003; Uyguner-Demirel; Bekbolet, 2011). This NOM fraction is environmentally significant and is known to be important for several environmental reactions and for the behavior of other chemical species. HS are considered recalcitrant compounds, if their resistance to chemical and microbiological degradation is taken into consideration, and they are formed by heterogeneous

mixing of organic aggregates.

       Photochemical reactions play a major role in the decay of natural organic compounds and the production of reactive species. These reactions are essential to the balance of ecological systems and carbon cycling, where dissolved organic carbon (DOC) is transformed into low-molecular weight organic compounds; the resulting organic compounds become more

bioavailable or undergo mineralization, generating a more labile fraction of NOM (Moran; Zepp, 1997; Paul, et al. 2004). Solar radiation in the presence of dissolved oxygen acts on the chromophoric dissolved organic matter (CDOM), producing reactive oxygen species as hydroxyl, peroxyl, hydroperoxyl groups and hydrogen peroxide; these species are important for the subsequent oxidation reactions of NOM and other compounds (O'Sullivan, et al. 2005; Paul,

et al. 2012).

       Some authors have differentiated aquatic NOM by lability or recalcitrance (Leenheer; Croué, 2003; Filella, 2009; Lindell; Granéli; Tranvik, 1995; Miller et al., 1997; Sleighter et al., 2014; Uyguner-Demirel; Bekbolet, 2011). Labile organic matter (LOM) is defined as a fraction of NOM that is more biodegradable. This bioavailable fraction is formed by autochthonous

production or by the photodegradation of organic compounds. Recalcitrant organic matter (ROM) is derived from allochthonous formations; it is less reactive to microbiological degradation in aquatic systems because it has already been transformed in soil; Thus, photochemical reactions are important for the transformation of NOM (Filella 2009; Dolgonosov and Gubernatorova 2010).

Hydrogen peroxide originating from aquatic environments, contributes to redox reactions and affects the dynamics of other chemical species, such as the oxidation of organic



compounds (Cooper, et al 1988; Zepp; Faust; Hoigné 1992; Burns et al 2012). This behavior is mainly due to the formation of hydroxyl radicals (HO•) through water photolysis, also known as the photo-Fenton reaction; these hydroxyl radicals are generated through the reaction of solar

radiation and ion Fe (II). Thus, hydrogen peroxide is a significant component in NOM degradation in aquatic system (Zepp; Gao 1998; Paul et al. 2012).

In an aquatic environment, NOM contents can be centrally identified through total organic carbon (TOC) analysis. This method is important, but cannot determine which fraction of the LOM is responsible for the main reactions in the aquatic environment. Currently, organic

composition characterizations of NOM are obtained with advanced analytical techniques such as $^{13}$C NMR and pyrolysis-GC-MS (Leenheer; Croué, 2003; Nebiosso; Piccolo, 2013). These analyses provide close estimates of NOM molecular composition, but are expensive, time intensive and require sophisticated equipment.

In this study, we hypothesize that LOM is fresh and reactive NOM that has recently

been introduced to aquatic environments and that hydrogen peroxide photogenerated in aquatic systems can facilitate reactions that oxidize some parts of NOM, transforming them into NOM oxidizing. Some authors have applied this approach to LOM in aquatic systems, where the environmental importance of LOM is associated with its scavenger capacity of other species, as hydrogen peroxide, influencing the redox conditions of the environment (Bisinoti, 2005;

Jardim et al., 2010). In this paper, the kinetics of hydrogen peroxide consumption were investigated as a method to distinguish between LOM and ROM.

## 2 Experimental Section
### 2.1 Materials

The reagents employed in this study were analytical reagents. Hydrogen peroxide 30 % (v/v) (Vetec), N,N-dyethyl-1,4-phenylenediamine sulfate (97 %), peroxidase (highly stabilized from horseradish, essentially salt-free, lyophilized powder, 200-300 units mg$^{-1}$ solid), dibasic potassium phosphate, monobasic potassium phosphate, lignin (alkali) and sodium pyruvate were purchased from Sigma-Aldrich. Standard fulvic acid from the Suwanne River was

purchased from the International Humic Substances Society (IHSS, USA). The solutions were prepared in ultrapure water (MilliQ).

### 2.2 Experimental design
### 2.2.1 Analytical determinations



Hydrogen peroxide quantification was performed using the DPD (N,N-diethyl-1,4-phenylenediamine) spectrophotometric method (Bader et al. 1988). Absorbance measurements were made with a 100 mm path length cuvette at 551 nm using a 1600 UV-Visible Spectrophotometer (Shimadzu, Japan). Total iron concentrations of the samples were quantified with the method 3500-Fe B method using a graphite furnace atomic absorption

spectrophotometer (GFAAS) with Zeeman background correction (Varian, model AA280Z, California, USA) (APHA, AWWA, WEF, 1998). The dissolved organic carbon (TOC) content was determined using a total organic carbon analyzer (Shimadzu TOC-VCSN).

### 2.2.2 Microcosms experiments

We applied the kinetics of hydrogen peroxide consumption as an indicator of labile and recalcitrant OM in freshwater samples. Some organic compounds were chosen to represent models of LOM and ROM. To evaluate the effect of laboratory lighting, microcosm experiments were conducted in the presence and in the absence of light.

        Fulvic acid, lignin, and sodium pyruvate were used as organic matter model compounds.

Each compound was employed at concentrations of 0.5, 1.0, 3.0 and 5.0 mg L$^{-1}$ in total organic carbon (TOC). All experiments were conducted against a microcosm control, in the absence of model compounds.

        The microcosm experiments were performed under constant stirring in jacketed glass reactors, and the temperature was kept constant at 19 °C. Each reactor was filled with 1.5 L of

ultrapure water and model compound solutions (0.5, 1.0, 3.0 and 5.0 mg C L$^{-1}$). The control experiment was filled only with ultrapure water. Then, hydrogen peroxide was added to each microcosm to obtain a final concentration of 7.1 µmol L$^{-1}$. During the experiments, pH was monitored and hydrogen peroxide was quantified at various time points, following Sect. 2.2.1 until it was completely consumed.


### 2.2.3 Data treatment

        The measurements of hydrogen peroxide consumption were applied to kinetic models to determine the reaction orders and half-lives. Then, we used a standard equation to accurately determine LOM concentrations as a function of half-lives.

### *2.2.4 Assessment of hydrogen peroxide consumption kinetics in freshwater samples*

        Considering the previously evaluated results, the best model compounds were determined to be fulvic acid and sodium pyruvate. The microcosm experiments were conducted





using freshwater samples collected from the Preto River (20° 48'40.94" S 49° 21'13.62" W).
The Preto River is located in São José do Rio Preto city, São Paulo state, Brazil. Each

microcosm sample received 1.5 L freshwater and was spiked with 0.5 and 5.0 mg L$^{-1}$ TOC of
pyruvate and fulvic acid. The microcosm controls were filled with only freshwater and ultrapure
water. Prior to being added to the control samples, the collected freshwater was filtered through
a 1.2 µm glass fiber membrane, followed by filtration through a 0.45 µm cellulose acetate
membrane. This procedure was employed to minimize microbiological degradation of the

NOM present in the samples. Hydrogen peroxide was added at the same concentrations of the
previous experiments and was quantified following the procedure described in 2.2.1.

### 2.2.5 Quantification of labile and recalcitrant organic matter in freshwater and determining the effects of seasonality

    Water samples from the Preto River were collected monthly for one year. These
experiments were conducted following procedures similar to those previously described (item
2.2.4), with the exception of adding organic model compounds to the samples.

### 3 Results and Discussion

### 3.1 Kinetics of hydrogen peroxide consumption in the presence of organic model compounds:
155        fulvic acid, lignin and sodium pyruvate

        Figures 1(a) and 1(b) show the consumption of hydrogen peroxide at different
concentrations of fulvic acid and lignin. For microcosms spiked with these two organic model
compounds, the kinetic behavior of hydrogen peroxide consumption was similar to the
microcosm control and followed the zero order law; the consumption of hydrogen peroxide was

virtually non-existent until 1400 minutes after start of the kinetic experiment. Table 1 shows
the disappearance rates and half-lives of hydrogen peroxide in the presence of all the organic
model compounds tested. The control experiment was a zero order reaction and had a rate of
1.21 x 10$^{-4}$ mol min$^{-1}$ and half-life of 68.8 hours.



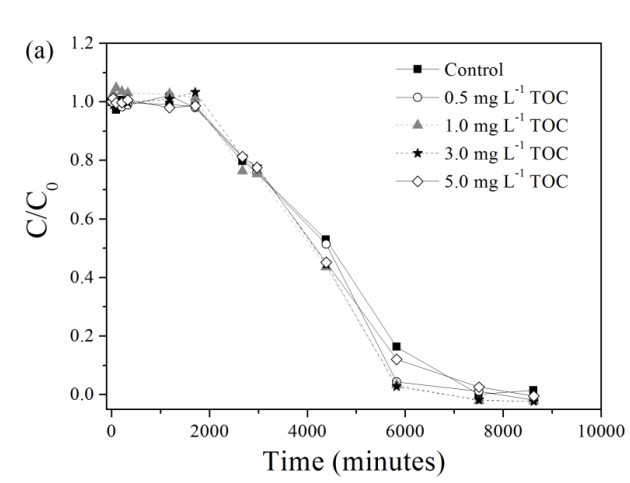

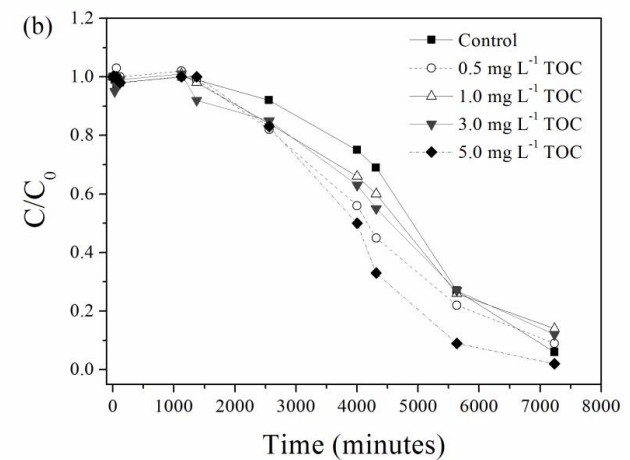

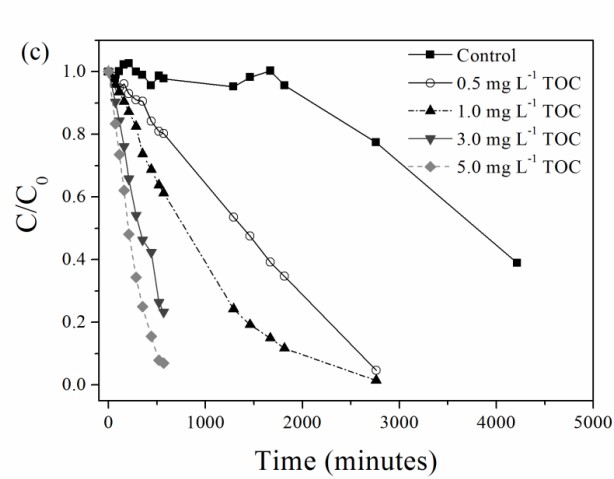

**Figure 1:** Kinetic monitoring of hydrogen peroxide (7.1 µmol L$^{-1}$) in the presence of organic matter models (0.5, 1.0, 3.0, 5.0 mg L$^{-1}$ TOC and control): (a) fulvic acid; (b) lignin and (c) pyruvate.

**Table 1:** Disappearance rate constants ($k$) of hydrogen peroxide (7.1 µmol L$^{-1}$) and half-life ($t_{1/2}$) in different concentrations (0.5, 1.0, 3.0 and 5.0 mg L$^{-1}$ TOC) of fulvic acid, lignin and pyruvate.

| Microcosms | Fulvic acid | | Lignin | | Pyruvate | |
|---|---|---|---|---|---|---|
| TOC (mg L$^{-1}$) | $k$ (µmol min$^{-1}$) | $t_{1/2}$ (hours) | $k$ (µmol min$^{-1}$) | $t_{1/2}$ (hours) | $k$ (min$^{-1}$) | $t_{1/2}$ (hours) |
| **Control** | $8.8 \times 10^{-4}$ | 69.2 | $7.98 \times 10^{-4}$ | 75.3 | $9.3 \times 10^{-5}$ | 89.2 |
| **0.5** | $8.5 \times 10^{-4}$ | 69.5 | $8.58 \times 10^{-4}$ | 68.1 | $5.3 \times 10^{-4}$ | 21.6 |
| **1.0** | $9.4 \times 10^{-4}$ | 64.9 | $7.75 \times 10^{-4}$ | 73.9 | $9.7 \times 10^{-4}$ | 11.9 |
| **3.0** | $10.9 \times 10^{-4}$ | 59.2 | $7.74 \times 10^{-4}$ | 71.7 | $2.2 \times 10^{-3}$ | 5.4 |
| **5.0** | $10.4 \times 10^{-4}$ | 59.6 | $8.79 \times 10^{-4}$ | 59.7 | $3.7 \times 10^{-3}$ | 3.1 |

* Values for triplicate experiments (CV < 14%)

The rates of hydrogen peroxide consumption in the presence of fulvic acid or lignin ranged from 8.6 to 10.9 x 10$^{-4}$ and 7.7 to 8.7 µmol min$^{-1}$, respectively (Table 1).

Different kinetic behavior was observed in the microcosms spiked with pyruvate (Fig. 1(c)). For this organic model compound, the kinetic profiles were different to what was observed for the control microcosm and showed that hydrogen peroxide consumption was influenced by the presence of pyruvate. This suggested that pyruvate was oxidizing and hydrogen peroxide was reducing, supporting the hypothesis of this work and confirming that LOM can be oxidized by hydrogen peroxide. In this experiment, hydrogen peroxide



consumption rates were faster, regardless of how much pyruvate was added to the microcosms. Rates ranged from $2.2 \times 10^{-3}$ to $9.7 \times 10^{-4}$ min$^{-1}$, and the kinetics of hydrogen peroxide consumption in presence of pyruvate were first order (Table 1).

In the presence of pyruvate at concentrations of 0.5, 1.0, 3.0 and 5.0 mg L$^{-1}$ of TOC, half-lives diminished by 75.8, 86.7, 93.9 and 96.5%, respectively, compared to the control
microcosms. Thus, we concluded that pyruvate was a suitable model compound for LOM, according to the kinetics of hydrogen peroxide consumption.

Our goal was to propose a method to quantify LOM in freshwater systems, based on half-lives of hydrogen peroxide, obtained in the experiment with pyruvate (see Table 1). To this end, we associated all the half-lives with their respective labile organic carbon concentrations,
which made it possible to develop a model equation that allowed us to calculate LOM concentrations. The same direct relation could not be made for ROM compounds because the half-lives of hydrogen peroxide in the microcosms carried out with fulvic acid and lignin were not distinguishable from the control microcosm and did not vary with the concentration ranges used for the model compounds.

We tested the pyruvate model mathematically to generate an analytical curve with a range of 0.25; 0.50; 1.0; 2.0; 3.0; 4.0 and 5.0 mg L$^{-1}$ TOC of pyruvate versus the half-lives of hydrogen peroxide consumption obtained in item 3.1 (Fig. 2).

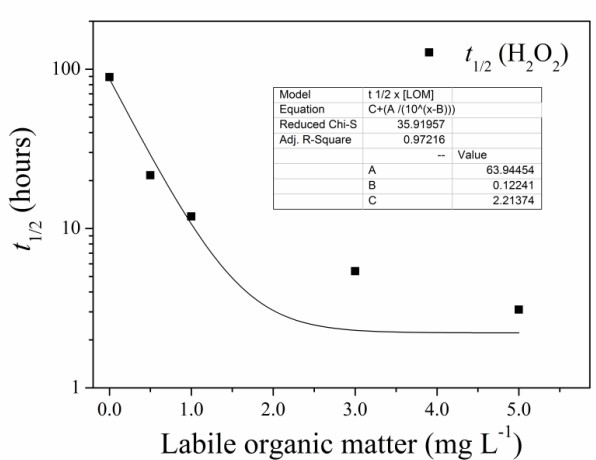

**Figure 2:** Hydrogen peroxide half-life versus LOM concentrations.

The equation model (Eq. 1) was the best fit for this objective (97% regression).

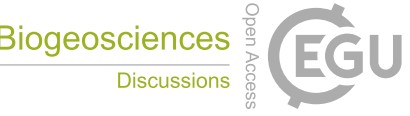

$$t_{1/2} = C + \left( \frac{A}{10^{([LOM]-B)}} \right) \tag{1}$$


Equation 1 was able to relate half-lives (hours) as a function of LOM (mg L$^{-1}$). The constants A and B are dependent on each other and are associated with curve inflection and the rate of hydrogen peroxide consumption. A and B were determined to be 63.9 and 0.12, respectively (Fig. 2). The constant C represents the minimum half-life value (2.2) at which

LOM concentration is influenced by hydrogen peroxide consumption. Quantification of LOM was necessary to determine the kinetics of hydrogen peroxide consumption in freshwater samples, following the procedures described in section 2.2.4, and to apply the half-lives obtained in Eq. 2.

$$[LOM] = \log \left( \frac{63.9}{t_{1/2} - 2.2} \right) + 0.12 \tag{2}$$

For this method, the quantification of LOM was suitable to $78.2 \leq$ half-life $> 2.2$ hours. In terms of LOM concentration, the equation was applicable to LOM concentrations of up to 2.9 mg L$^{-1}$.

ROM quantities were determined by applying the difference between TOC and LOM concentrations (Eq. 3).

$$ROM = TOC - LOM \tag{3}$$

### 230 3.2 Matrix effect assessment: Application of kinetic of hydrogen peroxide consumption to determine lability in freshwater samples

This experiment utilized freshwater samples from the Preto river and used the labile and recalcitrant organic matter model compounds (i.e., pyruvate and fulvic acid), following the previously obtained results. We assembled another experiment to evaluate the kinetics of

hydrogen peroxide consumption to distinguish and quantify the labile and recalcitrant fractions of organic matter in the freshwater samples. The kinetic profile of hydrogen peroxide disappearance in the freshwater samples spiked with the addition of the model compounds, and the plot is shown in Fig. 3. For all microcosms from the freshwater samples, with or without





the addition of organic matter, the kinetics of consumption were faster and distinguishable from
the control microcosm with ultrapure water.

Different kinetic behaviors of hydrogen peroxide consumption were observed for the freshwater microcosms that were and were not filtered (see Fig. 3). The disappearance of hydrogen peroxide was faster for the sample that was not filtered. This behavior can be attributed to the microorganisms present in the freshwater samples or the particulate material.
Some authors assessed these biological effects on the disappearance of hydrogen peroxide in coastal and offshore water samples. The results showed that filtration removed particulate material (minerals, organic components, biota) from the water samples and slowed down the rate of hydrogen peroxide consumption; initial sample concentrations were maintained for 30 hours, followed by an increase in consumption rates (Pestane and Zika, 1997). These results are
in agreement with the zero order results obtained in this work.

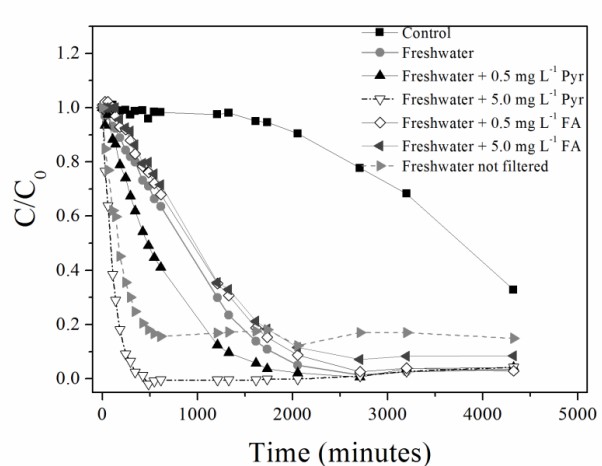

**Figure 3:** Disappearance of hydrogen peroxide in freshwater samples (filtered and not filtered) spiked (0.5 and 5.0 mg L$^{-1}$ TOC) with fulvic acid (FA) and pyruvate (Pyr).


The freshwater samples spiked with fulvic acid (0.5 and 5.0 mg L$^{-1}$ TOC) had hydrogen peroxide consumption profiles very similar to microcosm experiments carried out only in freshwater, which had rates between $3.1 \times 10^{-3}$ and $2.7 \times 10^{-3}$ μmol min$^{-1}$ and half-lives of between 18.9 and 19.6 hours. The microcosm with only freshwater had a faster rate of $3.8 \times 10^{-3}$
μmol min$^{-1}$ and a half-life of 15.8 hours (Table 2). These results indicate that freshwater from the Preto River predominantly consists of ROM.





The microcosms with freshwater samples spiked with pyruvate (0.5 and 5.0 mg L$^{-1}$ TOC) had rates and half-lives of $1.46 \times 10^{-3}$ min$^{-1}$ and 7.9 hours and $8.81 \times 10^{-3}$ min$^{-1}$ and 1.3 hours, respectively.


**Table 2:** Disappearance rate constants ($k$) of hydrogen peroxide (7.1 µmol L$^{-1}$) and half-life ($t_{1/2}$) in freshwater samples without and spiked MO models 0.5 and 5.0 mg L$^{-1}$ TOC of fulvic acid and pyruvate.

| Microcosms | $t_{1/2}$ (H$_2$O$_2$) (hours) | Reaction order | $k_{H2O2}$ |
|---|---|---|---|
| Freshwater Preto River | 15.8 | Zero Order | $3.8 \times 10^{-3}$ µmol$^{-1}$ min$^{-1}$ |
| Freshwater Preto River + 0.5 mg L$^{-1}$ Pyr | 7.9 | 1$^{st}$ Order | $1.5 \times 10^{-3}$ min$^{-1}$ |
| Freshwater Preto River + 5.0 mg L$^{-1}$ Pyr | 1.3 | 1$^{st}$ Order | $8.8 \times 10^{-3}$ min$^{-1}$ |
| Freshwater Preto River + 0.5 mg L$^{-1}$ FA | 18.9 | Zero Order | $3.1 \times 10^{-3}$ µmol$^{-1}$ min$^{-1}$ |
| Freshwater Preto River + 5.0 mg L$^{-1}$ FA | 19.6 | Zero Order | $2.7 \times 10^{-3}$ µmol$^{-1}$ min$^{-1}$ |
| Freshwater Preto River (not filtered) | 2.9 | 1$^{st}$ Order | $3.9 \times 10^{-3}$ min$^{-1}$ |

270       This highlights how the highest pyruvate concentrations resulted in the lowest half-life. In this experiment, increments of 0.5 to 5.0 mg L$^{-1}$ TOC (pyruvate) in freshwater samples resulted in an 83.5 % decrease in the half-lives obtained (Table 2). This same behavior was obtained in the experiment with ultrapure water, and resulted in a decrease in half-life of 85.5 % (Table 1). These results allowed us to conclude that pyruvate could be used as model of LOM
in freshwater samples without disturbing matrix effects.

Iron concentrations were below 10.0 µg L$^{-1}$ in all experiments that used ultrapure or freshwater samples.

### 3.3 Quantification of NOM labile and recalcitrant in freshwater sample: Approaches of
280       seasonality effects

The kinetic study of hydrogen peroxide consumption in freshwater samples was carried out over 12 months to evaluate lability variations in aquatic NOM as a function of seasonality such as during rainy and dry periods. The hydrogen peroxide half-lives were calculated monthly in the freshwater samples from the Preto River (see Fig. 4).

285       In the region of study, the hydrological cycle consists of two distinct periods. The rainy season spans from October to March, while dry season consists of the period between May and September. Sampling for the experiments was always performed at least three days after the last day of rain. The TOC average in the samples was 2.4 mg L$^{-1}$ for the year. The results





obtained in this experiment did not determine differences resulting from the presence or absence

of light (p ≤ 0.05).

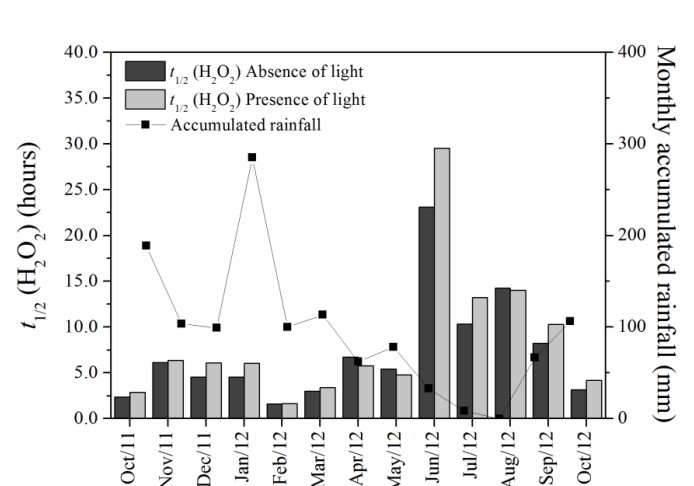

**Figure 4:** Effects of seasonality: Monthly accumulated rainfall and half-life of hydrogen peroxide in freshwater samples.


Figure 4 demonstrates how the rainy season generated samples with lower half-lives than those of the dry season. From October (2011) to March (2012), there was a period of higher rainfall where the half-lives average ranged from 1.6 to 6.2 hours; in contrast, the recorded half-lives were between 11.8 to 26.3 hours during the dry season (June to August). These results

demonstrate the influence of seasonality. A similar behavior of hydrogen peroxide disappearance in freshwater samples collected from the Amazon region has previously been observed and they noted slower hydrogen peroxide consumption rates in freshwater collected during the dry season than in the water samples that were spiked with LOM (low degraded) from this region (Jardim et al, 2010).

During the rainy season, there is major runoff of soil and this phenomenon delivers fresh NOM inputs to aquatic environments (Jardim et al., 2010; Westhorpe and Mitrovic 2012). These NOM inputs are fresher and more labile; owing to their short residence times in soil they are less likely to undergo chemical or microbiological processes. Similar effects were demonstrated by Sleighter et al (2014), higher DOM biolabile concentrations were observed to the season of

leaf fall due to increasing fresh DOM content in the leached. Biodegradation models of NOM in aquatic environments showed that the degradation times of lignin residues to low-molecular




compounds were shorter than decay times for lignin macromolecules originating from humification processes (Dolgonosov and Gubernatorova, 2010). On the other hand, runoff events are less impacting, and less NOM originates from the soil during the dry period; longer

residence times of NOMs in this environment make them more available for degradation processes, such as humification and aggregation. Consequently, when this organic material leaches into the aquatic environment, it is more degraded than NOM that leaches during the rainy season. Thus, we concluded that seasonality is related to the lability and recalcitrance of NOM.

The amounts of LOM and ROM were calculated using Eq. 2 and 3 (considering the average LOM and ROM obtained in the light and dark experiments). During the rainy season, LOM concentrations were higher for samples collected after a month with higher rainfall. October (2011), March and April (2012) had LOM concentrations of 76.8, 88.4 and 57.6 %, respectively (see Fig. 5). We note that there were some rainfall events in every month, so the

presence of LOM and ROM was expected.

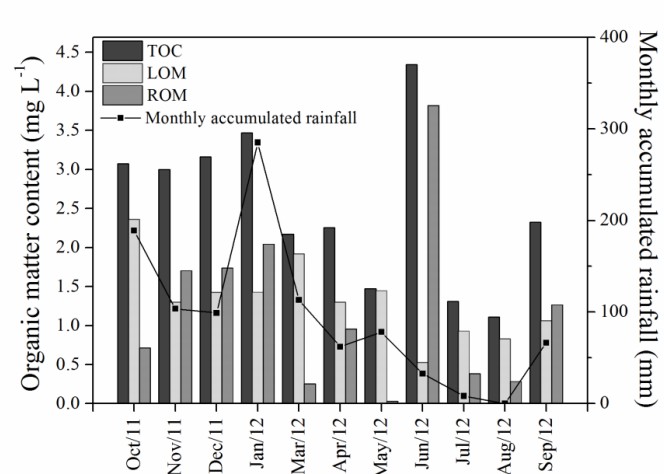

**Figure 5:** Total amounts of organic carbon (TOC), labile organic matter (LOM) and recalcitrant organic matter (ROM) (mg L$^{-1}$) in freshwater samples.


A study employed pyrolysis-GC-MS on samples collected in the same region as those used in this study and determined the molecular composition of aquatic humic substances (AHS) extracted from the Preto River (dry and rainy season). The authors determined that the composition of AHS differed between the rainy and dry seasons. Higher contents of aromatic



compounds, which were derived from lignin; were found during the dry season. These entities were leached from soil runoff and were related to the use and occupation of the surrounding land (Tadini et al, 2015). These data agree with the behavior that was observed in the kinetics of hydrogen peroxide consumption in freshwater samples and result from higher recalcitrant (aromatic) content in the samples during the dry season.

**Conclusion**

We concluded that the method proposed in this paper is an advance for environmental studies and can be employed as a new tool to determine the concentrations of labile and recalcitrant OM in freshwater. LOM concentrations can be quantified from 0.0 to 2.9 mg L$^{-1}$. ROM concentrations were obtained by determining the difference between TOC and LOM in

aquatic environments, which means there are no range restrictions when quantifying concentrations. This simple method was applied to water samples collected monthly over one year; LOM concentrations ranged from 0.47 to 2.1 mg L$^{-1}$, and ROM concentrations ranged from 0.08 to 3.5 mg L$^{-1}$. The results fell into expected ranges according to seasonality, where rainy periods produced higher quantities of LOM in aquatic environments due to runoff events.

Furthermore, this method is a great improvement to existing environmental methods because advanced analytical techniques are not necessary for simple characterizations on the degree of LOM present in freshwater samples.

**Acknowledgments**

This work was supported by a scholarship and sponsor from Foundation for Research Support of the State of São Paulo (FAPESP) (Processes 2012/06403-7, 2012/23066-4 and 2015/22954-1) and National Council for Scientific and Technological Development (CNPq).

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
