# Peer review of "Lability of natural organic matter in freshwater: a simple method for detection using hydrogen peroxide as an indicator"

_Biogeosciences, 2018_

## Referee Comment (RC1) · Anonymous Referee #1 · 27 Mar 2018

General comments:

The method presented in this paper appears to have potential to be a useful method for quantifying labile organic matter in environmental water samples. The quantification of labile organic matter can provide very meaningful information about the biogeochemical conditions in a particular environment and is useful across a wide range of scientific studies. However, the presentation of the method and interpretation of results in this paper are generally unclear and in places the interpretations appear flawed (more on this later). Furthermore, many of the decisions/approaches made by the authors lack sufficient justification/support and thus it is not possible to assess the validity of these

decisions. The authors should also provide more background on other methods of quantifying labile organic matter (e.g. incubations with native microbes) and the advantages and disadvantages of their method compared to existing ones. I think there may be some real advantages to their method but these need to be discussed by the authors. In addition, it is not made completely clear that the method the authors present here is new. I am under the impression that it is, but if it is in fact completely new they should state that more clearly. If the method is not new then they should make it clear how they have improved upon previous work. The authors should also discuss exactly what the method is measuring and how the model organic substances used (in particular pyruvate) compare to labile organic matter (more on this below). In particular, we see that pyruvate is more labile than the other model compounds (fulvic acid and lignin) though this in no way implies that pyruvate is a good model for LOM. It may be true that pyruvate is a reasonable model for LOM, however the authors need to explain what constitutes a "good" model for LOM and why pyruvate meets these criteria. Overall, I feel that the method described in the paper may provide a useful approach to quantifying labile organic matter. However, the paper currently lacks clarity and provides insufficient justification/support to allow for a full assessment of the significance of this work. As the paper currently stands there are a number of very significant issues that need to be addressed. The paper will require substantial revisions and reworking to allow for a full assessment of the potential scientific contributions.

Specific comments:

Note: Numbers in parenthesis are line numbers

(42) "...and the biogeochemical processes involved" Involved in what? Are you referring to the processes generation NOM?

(44) "According to some studies nearly 80% of NOM is composed of recalcitrant fractions..." Is this a general consensus or only something that a few studies agree upon?

(47) "important for several environmental reactions..." such as?

(48-49) "HS are considered recalcitrant compounds if their resistance to chemical and microbiological degradation is taken into consideration" (my emphasis added). Isn't this how recalcitrant is defined? You might want to change "if their resistance" to "as they are resistant to" and remove "is taken into consideration". This would make more sense.

(63-64) "...that is more biodegradable". More relative to what? Maybe rewrite as "LOM is operationally defined as the fraction of NOM that is biodegradable under a set of defined conditions"

(64-65) "...by the photodegradation of organic compounds". Provide citations here.

(65-67) Is it always (or generally) true that allochthonous material is more recalcitrant? If so provide citations.

(70-76) This paragraph is awkwardly written and unclear.

(77-79) These sentences are awkward/unclear. Maybe say something like "NOM can be measured by (state method), however this method only provide information about bulk concentration and does not provide info about the relative amounts of LOM and ROM"

(84-87) Very unclear. You write "we hypothesize that LOM is fresh and reactive..." By definition LOM is reactive so as stated it is an awkward hypothesis and needs to be reworded or reworked.

(86-87) "...transforming them into NOM oxidizing." This sentence does not make sense.

Section 2.2.2 Microcosms experiments is not clearly presented. It is not obvious from reading whether you are describing the method that is then used in section 2.2.4 or if you are describing a separate set of experiment from section 2.2.4. You should make

this more clear by possibly renaming section 2.2.2 to "Microcosm experiments: Model organic compounds".

(116-117) "Some organic compounds were chosen to represent models of LOM and ROM." Which compounds for LOM and which for ROM? I realize that pyruvate is for LOM and the others for ROM but it is awkward to write "some organic compounds" as opposed to stating which ones. Also you should provide a discussion justifying why you chose these compounds as model compounds. You obviously had some reason for selecting these but need to justify the decision and provide citations supporting the decision where applicable.

Section 2.2.3 Data treatment. There are essentially no details provided about the kinetic models used and how you fit your data to these models. You need to provide more details and justification of your approach.

(136) "Considering the previously evaluated results", what results are you referring to?

(136-137) "the best model compounds were determined to be fulvic acid and sodium pyruvate". Best how? What was your criteria for best? Presumably fulvic acid and lignin were tested to see which was a better ROM model. However, you only tested one potential LOM (i.e. pyruvate) so it is not fair to say it was the best model. You could potentially say it was a suitable model. This raises a more general issue, that there needs to be better justification/support for your choices. In particular, pyruvate appear (based on your data) to be more labile that fulvic acid. This is reasonable, but it does not necessarily mean that it is a good model for LOM. It is conceivable that another model compound may be even more labile than pyruvate. Thus, while pyruvate may be labile, it might not be nearly as labile as other compounds. Therefore, when you use pyruvate data in your kinetic modeling it becomes unclear exactly what your quantification of LOM means. More discussion and justification is needed here.

(143) Any reason why 0.45 um filter was used? A 0.2 um filter would have been more ideal for removing microbes. This may not really have affected your results, but you

should explain/justify your choice here.

(156-159) You state that the data follow a zero-order law. Is there any theoretical reason why this would be true? Even if the data follow this law, you should provide some discussion/justification as to why it follows a zero-order law.

Section 3.1. You state that H2O2 consumption is virtually non-existent until 1400 minutes after the start of the experiment. Why is this the case? You should provide conceptual/theoretical discussion explaining this.

(163) the numbers reported here do not agree with the numbers in Table 1.

Figure 1. The data do not all appear to follow a zero-order law. If the data fit a zero-order law, then C/Co vs. time should be linear (at least the portion following the 1400 minute lag should be). However, the data do not appear linear in many cases and thus your contention that they follow a zero-order relationship does not appear to be completely reasonable. This issue is part of a broader issue here where there is often insufficient justification for the statements made in the paper. As the paper is currently presented you do not explain how model parameters were fit (e.g., was a least squares fitting approach used) or how models were chosen. For example, was a zero-order model used because it fit better than a 1st-order model? Or was the model chosen for theoretical considerations. Discussion on these issues is required.

Table 1: Similar comments as for Figure 1. Also the number of significant figures for the "Lignin" column" differs from the other columns. Also you should include discussion and information showing how good the fits are for the estimated parameters (K). Currently there is no way to assess if the parameters reported in Table 1 are good fits. This is very important as it is presently not possible to assess if the kinetic models chosen to fit the experimental data are reasonable models. As I have mentioned above the zero-order fit for fulvic acid and lignin appears that it might not be all that reasonable. Furthermore, you should provide any available justification as to why the pyruvate data should fit a 1st-order model.

In section 3.1 you mention that pyruvate was oxidizing (as indicated by consumption of $H_2O_2$). However, you provide no indication of the extent of oxidation (did it completely oxidize to $CO_2$ or did it go to an simpler organic compound)? Did you measure changes in TOC during these experiments? If so it would be useful to include and discuss this data for both the pyruvate and fulvic acid experiments. This issue comes back to the previously stated issue that you do not explain exactly what you are measuring by using pyruvate as a model for LOM. It is totally conceivable that other model substances for LOM might oxidize more (and thus consume more $H_2O_2$) or less (and consume less $H_2O_2$) than pyruvate. Thus, had you used those substances you would have gotten completely different rate constants and your equations 2 and 3, which you use to estimate LOM in natural samples would have been different. A full and discussion of these issues, and presentation of data that might help to resolve this questions is crucial to demonstrating the utility of the method presented in your paper. Resolving/addressing these issues is crucial to demonstrating the validity/utility of your method.

(178) The numbers here do not agree with the table.

Equation 1: Justification for choosing this model should be discussed.

(249-250) You state that the results are in agreement with a zero-order model. If there was LOC in the samples wouldn't you expect (at least based on your earlier conclusions) that the samples would follow a 1st-order model. Recall that you stated your pyruvate data followed a 1st-order model and your fulvic data a zero-order model.

(255-275) Your conclusions/statements here are unclear and do not seem valid. You conclude that "these results indicate that freshwater from the Preto River predominantly consists of ROM". I would expect the freshwater only experiment to have very similar behavior to the freshwater with fulvic acid (since the fulvic acid adds only ROM). I would also expect the freshwater+pyruvate samples to consume $H_2O_2$ faster than the only freshwater sample, since the addition of pyruvate adds LOM. Thus, I do not believe that you can conclude from these results that the Preto River water consists predominantly of ROM based on your experimental results. In fact if the Preto River water was predominantly composed of ROM, then wouldn't you expect the data from the freshwater only experiment to be very similar to the control (Figure 3)? Since the freshwater only data is very different from the control, then I do not believe you can make the conclusions that you have made here. Thus, this section is very unclear and it is not obvious what exactly you mean to show with the data in Figure 3 and Table 2. Again, you will also need to justify the model choices in Table 2 and provide goodness-of-fit data/discussion for the K values estimated in Table 3.

(300-301) You mention that similar behavior of H2O2 consumption has been observed in Jardim et al. 2010. Was the same or similar method used in this paper? If so is your method new/modified? If Jardim et al. (2010) were doing something different from your current paper, then please make this clear.

---

## Author Comment (AC1) · 19 Apr 2018

We appreciate the valuable time and critical review done by referee 1 and certainly considered the useful comments and suggestions made to improve the manuscript. Please, find attached our answer for specific questions given in point by point as a supplement. Nevertheless, we would like to clarify the main points featured by the referee 1. The method proposed in this paper is new. We based on our experimental approach in previous observations published by Jardim et al (2010), in which it was suggested that $H_2O_2$ could be used to distinguish the difference between organic matter incorporated in waters during flooding periods in Negro River (Amazon Basin),

but it was not possible to quantify the amount of LOM. These authors used H2O2 kinetic consumption in two samples (freshwater from Negro River and water fortified with fresh leached soil organic matter). They showed a significant change in the chemical speciation of Hg coordinated by redox conditions in aquatic region studied in the presence of labile organic matter (LOM). In the rainy season, there was a great input of allochthonous natural organic matter (NOM) in aquatic bodies, and this NOM, considered fresh and reactive, would be able to scavenge H2O2 naturally photogenerated in the water column, influencing directly the oxidation conditions in this environment. Thus, this comprises one of the direct effects caused by the presence of LOM. In this work, we aimed at the possibility of quantifying labile and recalcitrant organic matter in freshwater samples. This objective was based on the importance that NOM plays in aquatic environment. It is known that NOM plays a relevant role in photoreactions, forming reactive species, or even scavenging these species. It is also primary source of biota and it is able to complex or adsorb other species as well. So, all these abilities are an intrinsic characteristic of NOM and its different reactivity degrees. Due to the complex composition of NOM, it is not feasible to carry out a characterization in molecular level as routine analysis. However, the information about the amount and the temporal variability of LOM in aquatic system would be very useful to explain many different processes in environmental studies. Here, we denominated LOM as NOM that was few oxidized or degraded and it is still able to react as a scavenger of oxidant species in aquatic systems, and it probably represents fresh organic matter input. On the other hand, recalcitrant organic matter (ROM) is the fraction that had already suffered oxidation, and it is less reactive towards oxidant species, such as H2O2. Our approach is different from classical methods used to distinguish organic matter degraded by microorganism or chemically, such as the ones used in the biochemical oxygen demand (BOD) and chemical oxygen demand (COD) measurements, respectively. These methods reflect an estimation of the amount of oxygen necessary to degrade organic content in an aquatic sample. Other approaches in literature are summarized by Filella (2009), but

all of them are considered bioassays, such as the ones used to measure the fraction of NOM, assimilable organic carbon (AOC) based on the measurements of growing biomass, and the biodegradable dissolved organic carbon (BDOC) is considered part of dissolved fraction of organic C able to be assimilated by heterotrophic microflora. There is another approach used by Laird and Scavic (1990), in which they measured labile dissolved organic carbon or matter (LDOC or LDOM) by bioassay, so again the lability has been considered the bioavailable fraction of NOM. In our attempt, we led the lability and recalcitrance concepts through the chemical approach, trying to reach a simpler approach than the protocols currently used to determine labile fraction, that consider it as biodegradable fraction of NOM, hence they always include bioassays. Reagents necessary to carry out our approach are quite simple and easily obtained. The time spent on application is also less than the necessary to use a biotic assay. So, we picked out some organic compounds to be tested as models of ROM and LOM, based on their molecular complexity and also on their natural presence in aquatic environments (pyruvate, lignin, ascorbic acid, hydroquinone and fulvic acid). In the specific comments, we added more information about the other tested compounds and why they were not considered good models, that is the reason why they were not presented in this manuscript (see answer of comment 16). Now, we agree that the term best model, used for us in the manuscript can be sustainable for other compounds tested. The results found for ascorbic acid and hydroquinone, other labile model compounds, can be inserted in the new version of the manuscript. For our matter, we simulated the acting of some model compounds, e.g. pyruvate, lignin and fulvic acid as scavengers of H2O2 in controlled system in the microcosm experiments. It is important to add that the kinetic models employed here to determine the order and consequently half-lives of H2O2 were based on the mathematical strict sense of the classical kinetic laws, and it was not our attempt with this experimental approach to discuss about any specific chemical mechanism behind these reactions, once our focus is to apply the mathematical formalism to general systems, such as natural aquatic samples. We are aware that this would be an interesting topic of discussion,

but for this purpose it would be necessary more experiments to establish the rates related not only to H2O2 loss, but also to the other species in the kinetic reactions. Some data treatment was included, as well as we could include all of them as a supplementary material (see answer to comment 21). Considering our definitions, the lignin and fulvic acid had behaved as recalcitrant compounds, because they did not affect the natural consumption of H2O2. In this case, the H2O2 consumption profile followed the same profile presented by control (ultrapure water), indicating that the kinetic of H2O2 decomposition was not affected by the presence of lignin, as well as of the fulvic acid. A distinct behavior in H2O2 kinetic consumption was observed when pyruvate was added. The kinetic consumption was faster as more pyruvate was spiked and we observed the great difference between these microcosms and the control. This meant that pyruvate plays a scavenger role in H2O2 consumption. Thus, we considered pyruvate a good and suitable LOM model. In the next step, we tried to find a correlation between the amount of pyruvate, here represented by TOC (denominated as LOM concentration, see Figure 2 on the manuscript), with the half-live times of H2O2 obtained in the kinetic experiment. We found an exponential equation provide the best fit for these data, leading us to define an equation that allowed to quantify LOM concentration from the half-live times obtained by H2O2 kinetic consumption. Then, we proposed the H2O2 kinetic consumption can be used to quantify LOM content in freshwater samples. We highlighted that organic model compounds considered as recalcitrant, were not used to define a way to quantify recalcitrant organic matter content. They were used to compared the scavenger effects caused by labile compound in H2O2 loss. Finally, to quantify recalcitrant organic matter concentration in freshwater samples, we suggested this can be calculated by the difference between TOC and LOM. After to define the equations to quantify LOM and ROM amount, we assessed possible matrix effects of environmental sample. For this, we carried out H2O2 kinetic consumption in freshwater collected of Preto river, and also, we did a standard addition of the model compounds, pyruvate (labile model) and fulvic acid (recalcitrant model) in this freshwater sample. Considering that

a presence of microorganisms in freshwater samples is ubiquitous, and they comprise a significant sink of H2O2, as related in literature (Pestane and Zika, 1997), we tested also a filtration step using 0.45 $\mu$m membrane, to exclude particulate fraction (inorganic and organic) and to minimize the influence of biota in H2O2 loss. For this experiment, profile of disappearance H2O2 was similar to filtered freshwater (only) and that one was spiked with fulvic acid (0.5 and 5.0 mg L-1). These results were expected, as we showed that fulvic acid had not or had few influence as a scavenger specie to H2O2. In the microcosms of freshwater filtered and spiked with pyruvate the kinetic of H2O2 consumption was faster as more pyruvate was added, showing the effect of increment of labile compound, a similar behavior observed in the microcosms conducted with pyruvate solutions in ultrapure water. So, we considered that there was not a matrix effect. Therefore, in this experiment, we did not expect to have a similar behavior only based in kinetic order of H2O2, as it was questioned (see response to comment 26). Since, using a real sample, we did not have a simple system compose by ultrapure water and organic model compounds, there is a mixture of organic compounds and our approach was developed to be sensitive to labile content. Further ahead, we have done this calculation to show that ROM content in this freshwater sample tested was higher than LOM. Beyond this, we applied this approach develop in the freshwater of Preto river during a year, to quantify LOM and ROM concentrations and also to related these amounts with seasonal effects. Finally, we would like to reinforce that we agree on the rewriting of some topics in this manuscript would be meaningful and this is feasible to carry out. Please, find attached our answer to specific comments and some suggestions to make some of these topics clearer in the supplement.

Please also note the supplement to this comment:
https://www.biogeosciences-discuss.net/bg-2018-122/bg-2018-122-AC1-supplement.pdf

―――――――――――――

**Supplement:**

April 16[th], 2018.

To: Biogeosciences – Discussion manuscript - bg-2018-122
Subject: Response to referee 1

We appreciate the valuable time and critical review done by referee 1 and certainly considered the useful comments and suggestions made to improve the manuscript. Please, find ahead our answers for specific questions given in point by point below. Nevertheless, first we would like to clarify the main points featured by the referee 1.

**General comments**

*The method presented in this paper appears to have potential to be a useful method for quantifying labile organic matter in environmental water samples. The quantification of labile organic matter can provide very meaningful information about the biogeochemical conditions in a particular environment and is useful across a wide range of scientific studies. However, the presentation of the method and interpretation of results in this paper are generally unclear and in places the interpretations appear flawed (more on this later). Furthermore, many of the decisions/approaches made by the authors lack sufficient justification/support and thus it is not possible to assess the validity of these decisions. The authors should also provide more background on other methods of quantifying labile organic matter (e.g. incubations with native microbes) and the advantages and disadvantages of their method compared to existing ones. I think there may be some real advantages to their method but these need to be discussed by the authors. In addition, it is not made completely clear that the method the authors present here is new. I am under the impression that it is, but if it is in fact completely new they should state that more clearly. If the method is not new then they should make it clear how they have improved upon previous work. The authors should also discuss exactly what the method is measuring and how the model organic substances used (in particular pyruvate) compare to labile organic matter (more on this below). In particular, we see that pyruvate is more labile than the other model compounds (fulvic acid and lignin) though this in no way implies that pyruvate is a good model for LOM. It may be true that pyruvate is a reasonable model for LOM, however the authors need to explain what constitutes a "good" model for LOM and why pyruvate meets these criteria. Overall, I feel that the method described in the paper may provide a useful approach to quantifying labile organic matter. However, the paper currently lacks clarity and provides insufficient justification/support to allow for a full assessment of the significance of this work. As the paper currently stands there are a number of very significant issues that need to be addressed. The paper will require substantial revisions and reworking to allow for a full assessment of the potential scientific contributions.*

**Answer to general comments**

We thank the referee for the opportunity to answer and rewrite some points of the manuscript in order to clarify the lacks appointed.

The method proposed in this paper is new. We based on our experimental approach in previous observations published by Jardim et al (2010), in which it was suggested that $H_2O_2$ could be used to distinguish the difference between organic matter incorporated in waters during flooding periods in Negro River (Amazon Basin), but it was not possible to quantify the amount of LOM. These authors used $H_2O_2$ kinetic consumption in two samples (freshwater from Negro River and water fortified with fresh leached soil organic matter). They showed a significant change in the chemical speciation of Hg coordinated by redox conditions in aquatic region studied in the presence of labile organic matter (LOM). In the rainy season, there was a great input of allochthonous natural organic matter (NOM) in aquatic bodies, and this NOM, considered fresh and reactive, would be able to scavenge $H_2O_2$ naturally photogenerated in the

water column, influencing directly the oxidation conditions in this environment. Thus, this comprises one of the direct effects caused by the presence of LOM.

In this work, we aimed at the possibility of quantifying labile and recalcitrant organic matter in freshwater samples. This objective was based on the importance that NOM plays in aquatic environment. It is known that NOM plays a relevant role in photoreactions, forming reactive species, or even scavenging these species. It is also primary source of biota and it is able to complex or adsorb other species as well. So, all these abilities are an intrinsic characteristic of NOM and its different reactivity degrees. Due to the complex composition of NOM, it is not feasible to carry out a characterization in molecular level as routine analysis. However, the information about the amount and the temporal variability of LOM in aquatic system would be very useful to explain many different processes in environmental studies.

Here, we denominated LOM as NOM that was few oxidized or degraded and it is still able to react as a scavenger of oxidant species in aquatic systems, and it probably represents fresh organic matter input. On the other hand, recalcitrant organic matter (ROM) is the fraction that had already suffered oxidation, and it is less reactive towards oxidant species, such as $H_2O_2$. Our approach is different from classical methods used to distinguish organic matter degraded by microorganism or chemically, such as the ones used in the biochemical oxygen demand (BOD) and chemical oxygen demand (COD) measurements, respectively. These methods reflect an estimation of the amount of oxygen necessary to degrade organic content in an aquatic sample. Other approaches in literature are summarized by Filella (2009), but all of them are considered bioassays, such as the ones used to measure the fraction of NOM, assimilable organic carbon (AOC) based on the measurements of growing biomass, and the biodegradable dissolved organic carbon (BDOC) is considered part of dissolved fraction of organic C able to be assimilated by heterotrophic microflora. There is another approach used by Laird and Scavic (1990), in which they measured labile dissolved organic carbon or matter (LDOC or LDOM) by bioassay, so again the lability has been considered the bioavailable fraction of NOM.

In our attempt, we led the lability and recalcitrance concepts through the chemical approach, trying to reach a simpler approach than the protocols currently used to determine labile fraction, that consider it as biodegradable fraction of NOM, hence they always include bioassays. Reagents necessary to carry out our approach are quite simple and easily obtained. The time spent on application is also less than the necessary to use a biotic assay.

So, we picked out some organic compounds to be tested as models of ROM and LOM, based on their molecular complexity and also on their natural presence in aquatic environments (pyruvate, lignin, ascorbic acid, hydroquinone and fulvic acid). In the specific comments, we added more information about the other tested compounds and why they were not considered good models, that is the reason why they were not presented in this manuscript (see answer of comment 16). Now, we agree that the term best model, used for us in the manuscript can be sustainable for other compounds tested. The results found for ascorbic acid and hydroquinone, other labile model compounds, can be inserted in the new version of the manuscript.

For our matter, we simulated the acting of some model compounds, e.g. pyruvate, lignin and fulvic acid as scavengers of H2O2 in controlled system in the microcosm experiments. It is important to add that the kinetic models employed here to determine the order and consequently half-lives of H2O2 were based on the mathematical strict sense of the classical kinetic laws, and it was not our attempt with this experimental approach to discuss about any specific chemical mechanism behind these reactions, once our focus is to apply the mathematical formalism to general systems, such as natural aquatic samples. We are aware that this would be an interesting topic of discussion, but for this purpose it would be necessary more experiments to establish the rates related not only to H2O2 loss, but also to the other species in the kinetic reactions. Some

data treatment was included, as well as we could include all of them as a supplementary material (see answer to comment 21).

Considering our definitions, the lignin and fulvic acid had behaved as recalcitrant compounds, because they did not affect the natural consumption of $H_2O_2$. In this case, the H2O2 consumption profile followed the same profile presented by control (ultrapure water), indicating that the kinetic of $H_2O_2$ decomposition was not affected by the presence of lignin, as well as of the fulvic acid. A distinct behavior in $H_2O_2$ kinetic consumption was observed when pyruvate was added. The kinetic consumption was faster as more pyruvate was spiked and we observed the great difference between these microcosms and the control. This meant that pyruvate plays a scavenger role in H2O2 consumption. Thus, we considered pyruvate a good and suitable LOM model.

In the next step, we tried to find a correlation between the amount of pyruvate, here represented by TOC (denominated as LOM concentration, see Figure 2 on the manuscript), with the half-live times of H2O2 obtained in the kinetic experiment. We found an exponential equation provide the best fit for these data, leading us to define an equation that allowed to quantify LOM concentration from the half-live times obtained by H2O2 kinetic consumption. Then, we proposed the H2O2 kinetic consumption can be used to quantify LOM content in freshwater samples. We highlighted that organic model compounds considered as recalcitrant, were not used to define a way to quantify recalcitrant organic matter content. They were used to compared the scavenger effects caused by labile compound in H2O2 loss. Finally, to quantify recalcitrant organic matter concentration in freshwater samples, we suggested this can be calculated by the difference between TOC and LOM.

After to define the equations to quantify LOM and ROM amount, we assessed possible matrix effects of environmental sample. For this, we carried out H2O2 kinetic consumption in freshwater collected of Preto river, and also, we did a standard addition of the model compounds, pyruvate (labile model) and fulvic acid (recalcitrant model) in this freshwater sample. Considering that a presence of microorganisms in freshwater samples is ubiquitous, and they comprise a significant sink of H2O2, as related in literature (Pestane and Zika, 1997), we tested also a filtration step using 0.45 μm membrane, to exclude particulate fraction (inorganic and organic) and to minimize the influence of biota in H2O2 loss.

For this experiment, profile of disappearance $H_2O_2$ was similar to filtered freshwater (only) and that one was spiked with fulvic acid (0.5 and 5.0 mg $L^{-1}$). These results were expected, as we showed that fulvic acid had not or had few influence as a scavenger specie to H2O2. In the microcosms of freshwater filtered and spiked with pyruvate the kinetic of H2O2 consumption was faster as more pyruvate was added, showing the effect of increment of labile compound, a similar behavior observed in the microcosms conducted with pyruvate solutions in ultrapure water. So, we considered that there was not a matrix effect. Therefore, in this experiment, we did not expect to have a similar behavior only based in kinetic order of H2O2, as it was questioned (see response to comment 26). Since, using a real sample, we did not have a simple system compose by ultrapure water and organic model compounds, there is a mixture of organic compounds and our approach was developed to be sensitive to labile content. Further ahead, we have done this calculation to show that ROM content in this freshwater sample tested was higher than LOM. Beyond this, we applied this approach develop in the freshwater of Preto river during a year, to quantify LOM and ROM concentrations and also to related these amounts with seasonal effects.

Finally, we would like to reinforce that we agree on the rewriting of some topics in this manuscript would be meaningful and this is feasible to carry out. Please, find below our answer to specific comments and some suggestions to make some of these topics clearer.

*Answers to specific comments*

**1)**(42) *"...and the biogeochemical processes involved" Involved in what? Are you referring to the processes generation NOM?*

The sentence was corrected to: "The chemical composition of NOM in aquatic environments is complex and variable, due to different sources of precursor material and the degradation processes involved"

**2)** (44) *"According to some studies nearly 80% of NOM is composed of recalcitrant fractions…" Is this a general consensus or only something that a few studies agree upon?*

Here, we considered a better explanation. Some authors attribute that significant part of NOM is comprised to the humic substances, which are considered a recalcitrant fraction NOM pool. Therefore, we changed the highlighted topic to:

"Part of the recalcitrant character of NOM in aquatic environments is given by the presence of humic substances."

**3)** (47) *"important for several environmental reactions…" such as?*

We agree that a reviewed topic about the play role of NOM in aquatic environment can be incremented here.

"(…) such as they play an important role in immobilization of other chemical entities in environment, as metals and organic compounds, consequently its availability (Cooper; Zika, 1983; Miller; Rose; Waite, 2009). HS are considered recalcitrant compounds as they are partially resistance to chemical and microbiological degradation. The Photo-Fenton reaction is an important sink of organic compounds and source of OH$^{\bullet}$ radicals (Southworth; Voelker, 2003).

In addition, this fraction of DOM as well as, other chromophore dissolved organic compounds are able to absorb UV radiation, interacting in the photochemical reactions. In this context, DOM can be important on formation and scavenging of reactive transient species ($H_2O_2$, OH$^{\bullet}$, $O_2^{\bullet-}$) in aquatic environment, therefore it is a relevant factor on the redox conditions and metal speciation (Copper; Zika, 1983; Moffett; Zika, 1983; Zhou; Mopper, 1990).

DOM photoexcited act on reduction of $O_2$ to form reactive species, as $O_2^{\bullet-}$ (superoxide radical), which disproportion of its conjugated ($HO_2^{\bullet}$) form $H_2O_2$. Photogeneration is the major via of $H_2O_2$ in aquatic system (Cooper; Zika, 1983; Scully; McQueen; Lean, 1996). Production of hydroxyl radical can arise from by direct photolysis of DOM. On the other hand, DOM also act as a scavenger of these reactive oxygen species, being oxidized in secondary reactions (Zeep; Gao, 1998).

**4)** (48-49) *"HS are considered recalcitrant compounds if their resistance to chemical and microbiological degradation is taken into consideration" (my emphasis added). Isn't this how recalcitrant is defined? You might want to change "if their resistance" to "as they are resistant to" and remove "is taken into consideration". This would make more sense.*

Now read: "HS are considered recalcitrant compounds as they are partially resistance to chemical and microbiological degradation (…)"

**5)** *(63-64) "…that is more biodegradable". More relative to what? Maybe rewrite as "LOM is operationally defined as the fraction of NOM that is biodegradable under a set of defined conditions*

We agree with this comment and in the new version of the manuscript we will use "*LOM is operationally defined as the fraction of NOM that is easy to degradable considering oxidizing species present naturally in fresh water, such as H2O2 and its precursors and ROM is defined a more oxidizing and probably ancient organic matter present in water*".

**6)** *(64-65) "...by the photodegradation of organic compounds". Provide citations here.*
Citations can be included according these references:
Lindell, M.J., Granéli, W., Tranvik, L.J., 1995. Enhanced bacterial growth in response to photochemical transformation of dissolved organic matter. Limnol Oceanogr. 40, 195-199.
Kieber,D. J.; Mcdaniel, J.; Mopper, K. 1989. Photochemical source of biological substrates in sea water: Implications for carbon cycling. Nature 341: 637-639.
Strome,D. J.; Miller, M.C. 1978. Photolytic changes in dissolved humic substances. Int. Ver. Theor. Angew. Limnol. Verh. 20: 1248-1254.
Xiao, M., Wu, F., Wang, L., Li, X., Huang, R. 2013. Investigation of low-molecular weight organic acids and their spatiotemporal variation characteristics in Hongfeng Lake, China. J Environ Sci. 25, 237-245.

**7)** *(65-67) Is it always (or generally) true that allochthonous material is more recalcitrant? If so provide citations.*

We agreed with the referee, that we could not generalize this, but the sentence was written, considering the context of introduction, according with the Review paper of Leenheer and Croué (2003).
According Leenheer and Croué in Characterizing dissolved aquatic organic matter. Environ Sci Tech. 37, 18A-26A, 2003.
(…) "DOM and soil humus have similar chemistries. Indeed, operationally defined humic substances typically compose about 50% of the DOM of an average river."
(…) "Most of the NOM is considered to be refractory to rapid biodegradation."
(…) "Surveys conducted on rivers in the United States and France showed that the BDOC content ranged from a few to about 40%. The BDOC content of rivers varies with the origin of the NOM. Autochthonous NOM, which is produced from macrophites, algae, and bacteria, is more biodegradable than allochthonous NOM, which has a pedogenic origin."

**8)** *(70-76) This paragraph is awkwardly written and unclear.*

This paragraph about the natural formation of H2O2 in aquatic system and its acting will be rewritten.

**9)** *(77-79) These sentences are awkward/unclear. Maybe say something like "NOM can be measured by (state method), however this method only provide information about bulk concentration and does not provide info about the relative amounts of LOM and ROM"*

This sentence will be rewritten as it was suggested.

**10)** *(84-87) Very unclear. You write "we hypothesize that LOM is fresh and reactive: ..." By definition LOM is reactive so as stated it is an awkward hypothesis and needs to be reworded or reworked.*

We are in accord with rewriting these paragraphs in order to clear up, as we have explained in the response of the general comment.

**11)** *(86-87) "...transforming them into NOM oxidizing." This sentence does not make sense.*

We agree that this sentence looked unclear. We meant that a labile fraction of NOM can be a scavenger of oxidant species, as H2O2, so it would assume/transform an oxidizing character. Thus, this sentence can be reformulated to:

"In this study, we denominated LOM as NOM that was few oxidized or degraded and it is still able to react as a scavenger of oxidant species in aquatic systems, and it probably represents fresh organic matter input. On the other hand, recalcitrant organic matter (ROM) is the fraction that had already suffered oxidation, and it is less reactive towards oxidant species, such as $H_2O_2$."

**12)** *Section 2.2.2 Microcosms experiments is not clearly presented. It is not obvious from reading whether you are describing the method that is then used in section 2.2.4 or if you are describing a separate set of experiment from section 2.2.4. You should make this more clear by possibly renaming section 2.2.2 to "Microcosm experiments: Model organic compounds".*

Description of experimental design was reworked to clarify the development of the method. Please, find below the reorganization of section 2.2.2, now it would be named 2.3, as well as previous and subsequent topics:

(…)

*2.1 Materials*

The content was kept the same.

*2.2 Analytical methods*

The content was kept the same.

*2.3 Method for quantification of labile and recalcitrant organic matter in freshwater samples*

Determination of organic compounds to be applied to a standard in the quantification of labile and recalcitrant organic matter were carried out in microcosms experiments. Fulvic acid, ascorbic acid, hydroquinone, lignin and sodium pyruvate were tested for organic matter model compounds (candidates to be used as standard of LOM or ROM). Each compound was employed at concentrations of 0.5, 1.0, 3.0 and 5.0 mg L-1 in total organic carbon (TOC). All experiments were conducted against microcosm control, in the absence of model compounds.

The microcosm experiments were performed under constant stirring in jacketed glass reactors, and the temperature was kept constant at 19 °C. Each reactor was filled with 1.5 L model compound solutions (0.5, 1.0, 3.0 and 5.0 mg C L$^{-1}$). The control experiment was filled only with ultrapure water. Then, hydrogen peroxide was added to each microcosm to obtain a final concentration of 7.1 µmol L-1. During the experiments, pH was monitored and hydrogen peroxide was quantified at various time points, following Sect. 2.2 until it was completely consumed. UV-Vis spectra were obtained in the time zero (defined immediately after added of hydrogen peroxide) and in the end time of kinetic.

*2.3.1 Assessment of matrix effect*

The microcosm experiments were conducted using freshwater samples collected from the Preto River (20° 48'40.94" S 49° 21'13.62" W). Each microcosm received 1.5 L freshwater

and was spiked with 0.5 and 5.0 mg L$^{-1}$ TOC of pyruvate and fulvic acid. The microcosm controls were filled with only ultrapure water. Hydrogen peroxide was added at the same concentrations of the previous experiments and was quantified following the procedure described in 2.2.

**2.3.2 Assessment of particulate effects**

Water samples were collected in same place mentioned in section 2.3.1. Each microcosm received 1.5 L of freshwater, being one was filled with *in natura* freshwater (no filtered) and for the other one, the freshwater sample was filtered through a 1.2 µm glass fiber membrane, followed by filtration through a 0.45 µm cellulose acetate membrane. These experiments were conducted following procedures similar to those previously described (item 2.3.1), with the exception of adding organic model compounds to the samples. Hydrogen peroxide was added at the same concentrations of the previous experiments and was quantified following the procedure described in 2.2.

**2.4 Data treatment**

The kinetic models employed here to determine the order and consequently half-lives of H2O2 were based on the mathematical strict sense of the classical kinetic laws (Table X).

Table X – Functions applied in the mathematical data treatment of H2O2 kinetic consumption to define the orders and the respective half-life time.

| Order | Equation | Half-life time |
|---|---|---|
| Zero | $y = (-k_{H2O2}\,x) + [H_2O_2]_0$ | $t_{1/2} = \dfrac{[H_2O_2]_0}{(2\,k_{H2O2})}$ |
| 1st | $y = [H_2O_2]_0 \exp(-k_{H2O2}\,x)$ | $t_{1/2} = \dfrac{\ln(2)}{(k_{H2O2})}$ |
| 2nd | $y = \dfrac{[H_2O_2]_0}{(1 + k_{H2O2}\,x\,[H_2O_2]_0)}$ | $t_{1/2} = \dfrac{1}{([H_2O_2]_0\,k_{H2O2})}$ |

Where: $k_{H2O2}$: rate (slope); x: time (minutes); y = [H$_2$O$_2$]; [H$_2$O$_2$]$_0$: initial concentration of H$_2$O$_2$; $t_{1/2}$: half-life time.

The data obtained from the measurements of hydrogen peroxide consumption were fitted using exponential decay functions with different exponential orders to determine the parameters related to the consumption rates. Then, we used the equation (Eq. X) below to correlate the half-lives as a function of LOM concentrations values.

$$y = C + \frac{A}{10^{(x-B)}}$$

(Eq. x)

*Where: y: half-life time of H$_2$O$_2$ (hours); x: [LOM] (mg L$^{-1}$); A, B and C: constants empirically defined.*

**2.5 Labile and recalcitrant organic matter in freshwater according to seasonality**

The Preto River is located in São José do Rio Preto city, São Paulo state, Brazil and is used for water supply, being one of the main important aquatic bodies in Turvo/Grande watershed. During a year water samples were monthly sampling of Preto river and determination of LOM and ROM concentrations were carried out. These experiments were carried out following

procedures similar to those previously described (item 2.3), with the exception of adding organic model compounds to the samples and they were conducted in the presence and absence of light from the laboratory. Hydrogen peroxide was added at the same concentrations of the previous experiments and was quantified following the procedure described in 2.2.

**13)** *(116-117) "Some organic compounds were chosen to represent models of LOM and ROM." Which compounds for LOM and which for ROM? I realize that pyruvate is for LOM and the others for ROM but it is awkward to write "some organic compounds" as opposed to stating which ones. Also you should provide a discussion justifying why you chose these compounds as model compounds. You obviously had some reason for selecting these but need to justify the decision and provide citations supporting the decision where applicable.*

We picked out some organic compounds to be tested as models of ROM and LOM, based on their molecular complexity and also on their natural presence in aquatic environments (pyruvate, lignin, ascorbic acid, hydroquinone and fulvic acid). Xiao and co-workers (2013) demonstrated the contribution of photodegradation NOM in the formation of Low Molecular weight organic acids (LMWOAs) (e.g. lactic, acetic, pyruvic, sorbic, oxalic acid) as products of photodegradation of NOM in aquatic system.

**14)** *Section 2.2.3 Data treatment. There are essentially no details provided about the kinetic models used and how you fit your data to these models. You need to provide more details and justification of your approach.*

It is important to add that kinetic models employed here to determine the order and consequently half-lives of $H_2O_2$ were based on the mathematical strict sense of the classical kinetic laws, as an exponential decay formula, as to fit the data, and it was not our attempt with this experimental approach discuss about any specific chemical mechanism behind these reactions, once our focus is to apply the mathematical formalism to general systems such as natural aquatic samples

He has chosen the order of the exponential function, for each case, using the correlation of the fit as the mandatory parameter, so we could decide the best order for each model compound used. This topic was rewritten (see response to comment 12).

**15)** *(136) "Considering the previously evaluated results", what results are you referring to?*

A reorganization of the experimental section 2.2.2 and 2.2.3 will make this topic clearer. Even though, here the statement highlighted it was referred to the results obtained in the section 2.2.2, where we tested the organic model compounds. We agree that this colocation was not proper in the experimental section. Please, see our attempt to improve the description in this section in the answer given in the comment 12.

**16)** *(136-137) "the best model compounds were determined to be fulvic acid and sodium pyruvate". Best how? What was your criteria for best? Presumably fulvic acid and lignin were tested to see which was a better ROM model. However, you only tested one potential LOM (i.e. pyruvate) so it is not fair to say it was the best model. You could potentially say it was a suitable model. This raises a more general issue, that there needs to be better justification/support for your choices. In particular, pyruvate appear (based on your data) to be more labile that fulvic acid. This is reasonable, but it does not necessarily mean that it is a good model for LOM. It is conceivable that another model compound may be even more labile than pyruvate. Thus, while pyruvate may be labile, it might not be nearly as labile as other compounds. Therefore, when*

*you use pyruvate data in your kinetic modeling it becomes unclear exactly what your quantification of LOM means. More discussion and justification is needed here.*

We agree with the referee and propose in a reviewed version of the manuscript, where we will present the five compounds tested and the reason for two compounds have not been used as a model (and also, they were not included in the first version). Then, we reinforce that pyruvate showed feasible to our goal. Please, find below a brief discussion about these other organic compounds tested (ascorbic acid and hydroquinone):

As mentioned (see answer to comment 13), for the labile compounds we choose some examples of low molecular weight organic compounds. Our first attempt was employing the ascorbic acid (AA) as model organic compound. Microcosm experiments were realized similarly to reported in the section 2.3 (e.g. control (ultrapure water), 0.5; 1.0; 3.0 and 5.0 mg L$^{-1}$ of TOC (ascorbic acid)). $H_2O_2$ profile during the kinetic experiment using AA as model compound is showed below in Figure S1.

Figure S1 – Hydrogen peroxide kinetic in the presence of ascorbic acid (0.5; 1.0; 3.0 and 5.0 mg L$^{-1}$ TOC).(experimental conditions: [H2O2] = 7.1 µmol L$^{-1}$; temperature: 19.0 °C; pH 6.6 – 6.2 (Control); pH 6.8 – 7.0 (0.5 mg L$^{-1}$); pH 6.7 – 7.0 (1.0 mg L$^{-1}$); pH 6.7 – 7.1 (3.0 mg L$^{-1}$) e pH 6.8 – 7.0 (5.0 mg L$^{-1}$)).

[Figure]

In this first attempt, for the control, 0.5; 1.0 mg L$^{-1}$ of TOC (AA) microcosms, it was possible to observe a decreasing of H2O2 concentration until 1320 minutes. Before this period, H2O2 kept constant. For the microcosms employing AA with 3.0 e 5.0 mg L$^{-1}$ TOC seems have been occur a suppression of analytical signal in the measurement of H2O2 until the initial time (zero) or a fast consumption of H2O2 added (Figure S1).

Reactive oxygen species formation, as $^{\bullet}$OH radicals and H2O2 by ascorbic acid were reported by Li and co-workers (2012). They suggested that ascorbate in aqueous solution (AscH$^-$) in the presence of $O_2$ can form $O_2^{\bullet-}$ (reaction 1), even though with addition of $H_2O_2$, would be formed $HO_2^{\bullet-}$ (reaction 2). The specie $O_2^{\bullet-}$, more unstable, disproportioned to form $H_2O_2$, a more stable specie (reaction 1) as well.

$$AscH\text{-} + O_2 \rightarrow {}^{\bullet}Asc + O_2^{\bullet-} \tag{1}$$

$$AscH\text{-} + H_2O_2 \rightarrow {}^{\bullet}Asc + HO_2^{\bullet-} \tag{2}$$
$$2\ O_2^{\bullet-} + 2H+ \rightarrow H_2O_2 + O_2 \tag{3}$$

We believe in the probability of $H_2O_2$ added, it has been consumed in the reactions of production radical species, justifying its rapid consumption and very low $H_2O_2$ concentrations

observed. After around 1300 minutes, the increase in $H_2O_2$ concentration (microcosms 3.0 and 5.0 mg $L^{-1}$ TOC of AA) could be justified by the production mediated by reactive species formed (reaction 3). In this way the mathematical treatment to determine the order of reaction and the half-life of this experiment were calculated from the time of 1320 minutes. Rates of $H_2O_2$ consumption and half-live times are showed in Table S1.

**Table S1:** Disappearance rate constants ($k$) of hydrogen peroxide (7.1 µmol $L^{-1}$) and half-life ($t_{1/2}$) in different concentrations (0.5, 1.0, 3.0 and 5.0 mg $L^{-1}$ TOC) of ascorbic acid.

| Microcosms | $t_{1/2}$ ($H_2O_2$) | Kinetic order | $k_{H2O2}$ |
|---|---|---|---|
| Control | 64.2 | Zero | 8,3 x $10^{-4}$ µmol $^{-1}$ min $^{-1}$ |
| 0.5 | 50.8 | $1^{st}$ | 2,3 x $10^{-4}$ min $^{-1}$ |
| 1.0 | 48.7 | $1^{st}$ | 2,4 x $10^{-4}$ min $^{-1}$ |
| 3.0 | 62.6 | Zero | 1,6 x $10^{-3}$ µmol $^{-1}$ min $^{-1}$ |
| 5.0 | 79.4 | Zero | 1,3 x $10^{-3}$ µmol $^{-1}$ min $^{-1}$ |

Microcosms spiked with 0.5 e 1.0 mg $L^{-1}$ TOC of AA kinetic of $H_2O_2$ followed $1^{st}$ order, and AA seems to be behaved as $H_2O_2$ scavenger, but for microcosms incremented with 3.0 e 5.0 mg $L^{-1}$ of TOC of AA, where seems to happened a production of $H_2O_2$, the kinetic was defined as zero order ($t_{1/2}$ ($H_2O_2$), 62.6 e 79.4 hours). These results corroborated with the hypothesis of $H_2O_2$ production mediated by AA, because after a fast production of $H_2O_2$, considering that had been a decreasing of AA concentration, there was not more AA enough to scavenger the oxidant species, and $H_2O_2$ in this system followed a kinetic of decomposition as the control (zero order, $t_{1/2}$ ($H_2O_2$) 64.2 hours). An additional experiment was conducted, without the initial $H_2O_2$ spikes, only with a dissolution of the same concentration range of AA used on previously experiment in ultrapure water, followed the $H_2O_2$ measurements (Figure 2S).

The increasing of $H_2O_2$ in the microcosms was observed and it was directly proportional with AA concentrations, and a decreasing of oxidant was verified until 1335 minutes (Figure 2S). We highlighted that experimental conditions related of the analytical method used to quantify $H_2O_2$ was verified also, such as pH of solutions, and any unusual factor was not observed, and the $H_2O_2$ concentration in the control kept below detection limit. Thus, we confirmed there was a production of $H_2O_2$ associated to presence of AA.

Figure 2S – Concentrations of $H_2O_2$ in the presence of AA (without $H_2O_2$ addition). (experimental conditions: temperature: 19,0 °C; pH 6.1 – 6.3 (Control); pH 6.9 – 6.3 (0.5 mg $L^{-1}$); pH 5.5 – 6.3 (1.0 mg $L^{-1}$); pH 5.7 – 6.3 (3.0 mg $L^{-1}$) and pH 5.1 – 6.3 (5.0 mg $L^{-1}$)).

[Figure]

Other organic compound tested was the Hydroquinone, but it has not behaved as a model compound, because it had caused some interference in H2O2 measurements (suppression of analytical signal. This effect was attributed to the presence of Hydroquinone based on the results obtained to the Control microcosm, which in the results were as expected.

*17) (143) Any reason why 0.45 um filter was used? A 0.2 um filter would have been more ideal for removing microbes. This may not really have affected your results, but you should explain/justify your choice here.*

We agree with this opinion, even we have tested this approach for other natural water samples used in sequence of another work (data not showed here). However, for this work, presented here, we decided to use 0.45 μm membrane to guaranteed at least the dissolved fraction of organic matter, avoiding particulate fraction.

*18) (156-159) You state that the data follow a zero-order law. Is there any theoretical reason why this would be true? Even if the data follow this law, you should provide some discussion/justification as to why it follows a zero-order law.*

Please, see the response to comment 21, which in we presented a more detailed discussion about data treatment and the orders defined in these experiments. We highlighted this information can be included in a rewritten version.

*19) Section 3.1. You state that H2O2 consumption is virtually non-existent until 1400 minutes after the start of the experiment. Why is this the case? You should provide conceptual/theoretical discussion explaining this. Some studies showed the half-life time of hydrogen peroxide range to*

Hydrogen peroxide is a secondary standard and it is known that this compound decomposes naturally over time, depending of temperature and solution concentration. In this study, the microcosm control was conducted only with ultrapure water and spiked with H2O2, so it represents decomposition of $H_2O_2$, without any or minimum (if we considered that this experiment was not autoclaved) biotic contribution. We emphasize that fulvic acid and lignin behaved as recalcitrant compounds, they did not perform a scavenger role in H2O2, as it was observed in the resemble in the kinetic consumption profile. We can insert more information about this topic:

In aquatic environment, the major sinks of $H_2O_2$ seem to be catalytic decomposition mediated by enzymes, metal transitions (Fe and Cu), photo-Fenton reaction for example (Zepp; Faust; Hoignè, 1992; Southworth; Voelker, 2003) and the direct photocatalysis. In freshwater system, $H_2O_2$ half-live time was few hours, and was attributed to action of microorganisms. For seawater, the half-live times take order of days. Pestane and Zika (1997) demonstrated biotic effects in $H_2O_2$ loss kinetic in seawater. For unfiltered samples results ranged to 12 (coastal waters) to 120 hours (surface water from open ocean), which in they corelated effects of DOM that it was higher in coastal areas (higher absorbance in 300 nm). $H_2O_2$ natural concentrations ranged to 1.24 to 2.42 x $10^{-7}$ mol $L^{-1}$.

Comparing the measures of $H_2O_2$ disappearance kinetic in filtered (0.2 μm), unfiltered, and autoclaved seawater samples, the $H_2O_2$ loss rates was greatly decrease by the remoting of biotic effects (biota, mineral and organic detritus) in filtered sample. The profile of $H_2O_2$ obtained was the same observed in this work, which in initially there is a little or no degradation of $H_2O_2$ and the losses start around 30 hours. For the filtered and autoclaved samples there is not virtually disappearance of $H_2O_2$ until 80 hours, showing that this behavior depends on biotic characteristics of aquatic system. In this study, microcosms control was prepared with ultrapure water (MilliQ) spiked with H2O2, so it showed the decomposition of $H_2O_2$, without any or minimum (if we considered that this experiment was not autoclaved) biotic contribution. The

microsms with organic recalcitrant model compounds solutions (e.g. fulvic acid and lignin) showed a similar behavior, indicating no scavenger acting in H2O2 loss.

*20) (163) the numbers reported here do not agree with the numbers in Table 1.*

We verified and this data was typed wrong. Please, consider the following corrections, but in general, interpretation and pattern of results were not modified, even with this misconception of typed. The value corrects for $k$ of control (pyruvate microcosm) is 6.71 x $10^{-4}$ µmol min$^{-1}$ ($[H_2O_2]_0$ = 7.18 µmol L$^{-1}$). You could confirm, considering the zero order attributed for the $H_2O_2$ kinetic consumption in the control sample, the half-life time was calculated by $t_{1/2}$ = $([H_2O_2]_0)/(2 \times k_{H2O2})$. Please, find below the corrections:
Section 3.1
(…)
Table 1 shows the disappearance rates and half-lives of hydrogen peroxide in the presence of all the organic model compounds tested.
(…)
**Figure 1**

**Table 1:** Disappearance rate constants (k) of hydrogen peroxide (7.1 µmol L$^{-1}$) and half-life ($t_{1/2}$) in different concentrations (0.5, 1.0, 3.0 and 5.0 mg L$^{-1}$ TOC) of fulvic acid, lignin and pyruvate.

| Microcosms TOC (mg L$^{-1}$) | Fulvic acid | | Lignin | | Pyruvate | |
|---|---|---|---|---|---|---|
| | $k$ (µmol min$^{-1}$) | $t_{1/2}$ (hours) | $k$ (µmol min$^{-1}$) | $t_{1/2}$ (hours) | $k$ (min$^{-1}$) | $t_{1/2}$ (hours) |
| Control | $8.8 \times 10^{-4}$ | 69.2 | $7.98 \times 10^{-4}$ | 75.3 | $6.71 \times 10^{-4}$ | 89.2 |
| 0.5 | $8.5 \times 10^{-4}$ | 69.5 | $8.58 \times 10^{-4}$ | 68.1 | $5.3 \times 10^{-4}$ | 21.6 |
| 1.0 | $9.4 \times 10^{-4}$ | 64.9 | $7.75 \times 10^{-4}$ | 73.9 | $9.7 \times 10^{-4}$ | 11.9 |
| 3.0 | $10.9 \times 10^{-4}$ | 59.2 | $7.74 \times 10^{-4}$ | 71.7 | $2.2 \times 10^{-3}$ | 5.4 |
| 5.0 | $10.4 \times 10^{-4}$ | 59.6 | $8.79 \times 10^{-4}$ | 59.7 | $3.7 \times 10^{-3}$ | 3.1 |

The rates of hydrogen peroxide consumption in the presence of fulvic acid or lignin ranged from 8.5 to 10.9 x $10^{-4}$ and 7.7 to 8.8 µmol min$^{-1}$, respectively (Table 1).

*21) Figure 1. The data do not all appear to follow a zero-order law. If the data fit a zero order law, then C/Co vs. time should be linear (at least the portion following the 1400 minute lag should be). However, the data do not appear linear in many cases and thus your contention that they follow a zero-order relationship does not appear to be completely reasonable. This issue is part of a broader issue here where there is often insufficient justification for the statements made in the paper. As the paper is currently presented you do not explain how model parameters were fit (e.g., was a least squares fitting approach used) or how models were chosen. For example, was a zero-order model used because it fit better than a 1st-order model? Or was the model chosen for theoretical considerations. Discussion on these issues is required.*

In case of microcosms experiments using lignin and fulvic acid, the $H_2O_2$ had a consumption the zero order as showed, which in there is a little or no scavenging effect of these organic model compound in the $H_2O_2$ loss. We could add as supplementary material, the graphics showing the application of these mathematical models used. For now, please find below some examples of the mathematical treatment applied to define the kinetic orders (best fits) and their respective regression and $k$ (slope) (Figure 3S).

Figure 3S - $H_2O_2$ disappearance applying kinetic laws in the microcosms experiments using organic model compounds (A) 0.5 mg $L^{-1}$ TOC Lignin; (B) 5.0 mg $L^{-1}$ TOC Lignin; (C) 0.5 mg $L^{-1}$ TOC Fulvic acid; (D) 5.0 mg $L^{-1}$ TOC Fulvic acid; (E) 0.5 mg $L^{-1}$ TOC Pyruvate; (F) 5.0 mg $L^{-1}$ TOC Pyruvate; Control (ultrapure water): (G) during Lignin experiment; (H) during Fulvic acid experiment and (I) during Pyruvate experiment.

[Figure]

For the pyruvate microcosms, specifically to 0.5 and 3.0 mg L-1 of TOC, the zero order showed slightly best fit, instead the other ones, which in the best fit was obtained for the first order. Even though, we assumed the first order for all pyruvate microcosms, because the difference between the both highlighted were not great, but for 1.0 and 5.0 mg TOC of pyruvate the first order fit was great better. All these data can be included in the supplementary material..

**22)** *Table 1: Similar comments as for Figure 1. Also the number of significant figures for the "Lignin" column" differs from the other columns. Also you should include discussion and information showing how good the fits are for the estimated parameters (K). Currently there is no way to assess if the parameters reported in Table 1 are good fits. This is very important as it is presently not possible to assess if the kinetic models chosen to fit the experimental data are reasonable models. As I have mentioned above the zero-order fit for fulvic acid and lignin appears that it might not be all that reasonable. Furthermore, you should provide any available justification as to why the pyruvate data should fit a 1st-order model.*

Please, consider the response to comment 21.

**23)** *In section 3.1 you mention that pyruvate was oxidizing (as indicated by consumption of H2O2). However, you provide no indication of the extent of oxidation (did it completely oxidize to CO2 or did it go to an simpler organic compound)? Did you measure changes in TOC during these experiments? If so it would be useful to include and discuss this data for both the pyruvate and fulvic acid experiments. This issue comes back to the previously stated issue that you do not explain exactly what you are measuring by using pyruvate as a model for LOM. It is totally conceivable that other model substances for LOM might oxidize more (and thus consume more H2O2) or less (and consume less H2O2) than pyruvate. Thus, had you used those substances you would have gotten completely different rate constants and your equations 2 and 3, which you use to estimate LOM in natural samples would have been different. A full and discussion of these issues, and presentation of data that might help to resolve this questions is crucial to*

*demonstrating the utility of the method presented in your paper. Resolving/addressing these issues is crucial to demonstrating the validity/utility of your method.*

We carried out TOC measurements during kinetic experiments, but significant differences were not observed. We believe that what might have happened in this case, it was not a mineralization of C content, but just chemical transformations, as can be observed in UV-Vis spectra results (Figure 3S). In biological conditions, pyruvate can decompose $H_2O_2$ to form acetate $H_2O$ and $CO_2$ (Varma; Devamanoharan; Morris, 1990). So, measurements of $CO_2$ would be useful. However, microcosms were conducted in glass-jacket reactors opened, so it was not possible to verify at least an increase in total inorganic carbon (TIC) by TOC analyze. Even with this lack, we assume that pyruvate play as a scavenger of $H_2O_2$, considering significant distinction of $H_2O_2$ loss between the control and the others microcosms carried out, as well as differences in the absorbance in the UV-Vis spectra obtained between the begin and the end of $H_2O_2$ kinetic consumption, this variation have not seen for the microcosms conducted with Lignin and FA, as organic model compounds.

Figure 3S – UV-Vis spectra of microcosms samples during $H_2O_2$ kinetic consumption (I = Initial and F = Final) in the presence of organic model compounds (0.5; 1.0; 3.0; 5.0 mg L$^{-1}$ TOC): (A) Lignin; (B) Fulvic acid; (C) Pyruvate.

[Figure]

**24)** *(178) The numbers here do not agree with the table.*

This mistake was verified. It was corrected as showed in the response to comment 19.

**25)** *Equation 1: Justification for choosing this model should be discussed.*

Our approach was empiric, using the half-live times obtained to $H_2O_2$ consumption in the presence of pyruvate against the respective LOM concentrations. We have followed using exponential decay functions, except for this case we use the base 10 antilogarithm for didactic purposes.

$$y = C + \frac{A}{10^{(x-B)}}$$

*(Eq. x)*

*y: half-life time of $H_2O_2$ (hours); x: [LOM] (mg $L^{-1}$); A, B and C: constants empirically defined.*

Please, see response to comment 12. We have added more information and rewritten this topic.

**26)** *(249-250) You state that the results are in agreement with a zero-order model. If there was LOC in the samples wouldn't you expect (at least based on your earlier conclusions) that the samples would follow a 1st-order model. Recall that you stated your pyruvate data followed a 1st-order model and your fulvic data a zero-order model.*

As we have showed previously, we assumed that pyruvate had a good behavior to simulate the kinetic consumption of H2O2 in the presence of scavenger, besides of the model proposed to estimate LOM concentration, we proposed a second approach, considering ROM content can be calculated by the difference between TOC and LOM concentrations. Accordingly, in this experiment our aim was apply the methodology proposed and verify the possible matrix effects of this freshwater sample. Therefore, we did not expect to have a similar behavior only based in kinetic order of H2O2. Since, using a real sample, we did not have a simple system as that one compose by ultrapure water and organic model compounds only, actually there is a mixture of compounds and our approach was developed to be sensitive to labile content. Further ahead, we have done this calculation to show that ROM content in this freshwater sample is higher than LOM.

**27)** *(255-275) Your conclusions/statements here are unclear and do not seem valid. You conclude that "these results indicate that freshwater from the Preto River predominantly consists of ROM". I would expect the freshwater only experiment to have very similar behavior to the freshwater with fulvic acid (since the fulvic acid adds only ROM). I would also expect the freshwater+pyruvate samples to consume H2O2 faster than the only freshwater sample, since the addition of pyruvate adds LOM. Thus, I do not believe that you can conclude from these results that the Preto River water consists predominantly of ROM based on your experimental results. In fact, if the Preto River water was predominantly composed of ROM, then wouldn't you expect the data from the freshwater only experiment to be very similar to the control (Figure 3)? Since the freshwater only data is very different from the control, then I do not believe you can make the conclusions that you have made here. Thus, this section is very unclear and it is not obvious what exactly you mean to show with the data in Figure 3 and Table 2. Again, you will also need to justify the model choices in Table 2 and provide goodness-of-fit data/discussion for the K values estimated in Table 3.*

We are sorry that this part was not clear in the original manuscript. So, we have revised the contents of this part. However, to clarify this point, considering that proposed of this work was to establish a possible way to quantify LOM, we did not agree that the behavior of this freshwater sample must be similar to experiments presented (microcosm ultrapure water + adding FA), even it has higher ROM content than LOM, as showed by our results.

When we apply FA (in this case, it is an example of natural and relatively common fraction of NOM naturally found), as organic model compound, we would like to demonstrated that a recalcitrant compound does not have a great scavenger effect on the $H_2O_2$, compared to labile compound, here represented by pyruvate.

Therefore, we agree with the presented results and reinforce that we did not expect to have a similar behavior only based in kinetic order of $H_2O_2$. Since, using a real sample, we have not had a simple system compose by ultrapure water and organic model compounds, there is a mixture (LOM + ROM), and our approach was developed to be sensitive to labile content. For ROM, we did not propose use the AF as model, even though, we were not finger out an equation to this, just to LOM. We proposed measurements of ROM content was calculated indirectly by difference between TOC and LOM.

The kinetic profile of $H_2O_2$ in the sample (just freshwater) and samples fortified with FA (0.5 and 5.0 mg $L^{-1}$) occurred by effects caused by LOM present in the natural freshwater. We reinforce, we did not expect that FA increase the half-life times, as showed, the FA was considered recalcitrant because it did not play a scavenger role in the $H_2O_2$ kinetic, in other words, we could say that it is not reactive forward $H_2O_2$, in these experimental conditions used.

In the comments about the data (half-life times, kinetic order…) we agree that we can increment this item, showing data treatment used and to enrich the discussion, as in the example given in to response comment 21.

**28)** *(300-301) You mention that similar behavior of H2O2 consumption has been observed in Jardim et al. 2010. Was the same or similar method used in this paper? If so is your method new/modified? If Jardim et al. (2010) were doing something different from your current paper, then please make this clear.*

As already mentioned in the response to the general comment, the work developed in this manuscript was different of the presented by Jardim et al (2010). We inspired in their findings, where they showed the effect of the input of fresh NOM, considered labile, in the control of redox conditions in their study conducted in Amazon region. They highlighted that in the high flowed season, there is an input of fresh NOM in aquatic system and in this period, this reactive fraction of OM can act as a scavenger od $H_2O_2$ photogenerated in aquatic system, influencing directly the redox conditions, thereafter altering the oxidation of $Hg^0$ to $Hg^{2+}$. Therefore, in their work, Jardim and co-workers showed $H_2O_2$ kinetic consumption only in two samples, and they have not proposed a method, as we are trying here. They highlighted that $H_2O_2$ might be used as indicator, but they did not provide a further discussion about how to do this for freshwater samples. Considering this fact, we decide to test $H_2O_2$ as an indicator of presence of LOM for freshwater samples, as well as to establish a method to quantify this reactive fraction of NOM. A reviewed topic about this could be incremented in the introduction.

We would be glad to respond to any further questions and comments that you may have.
Thank you.
Sincerely yours,
All the authors.

---

## Referee Comment (RC2) · J. Hemingway (Referee) · 26 Apr 2018

The central focus of this manuscript is to describe a simple assay that utilizes the decay rate of hydrogen peroxide when exposed to natural organic matter (NOM) as a metric to quantify the amount of "labile" material present. This study first measures hydrogen peroxide decay rates in the presence of three laboratory reference materials (lignin, fluvic acid, and pyruvate) at multiple concentrations. Then, using the observation that pyruvate addition enhances hydrogen peroxide decay rate, thus lowering the half-life, the authors generate a model to predict the absolute concentration of "labile" NOM present in a sample, assuming that all "labile" material behaves identically to pyruvate.

All remaining NOM is considered "recalcitrant." This assay is then applied natural samples that have been enriched with the three laboratory reference materials, as well as time-series samples collected from a nearby river.

I have a number of issues with this study, beginning with the overall experimental design and the separation of NOM into "labile" and "recalcitrant" pools. As I understand it, the logic of this experimentation is as follows: (i) choose three reference materials to see if HOOH decays more quickly in their presence, (ii) observe this to be true in the case of pyruvate, (iii) quantify HOOH decay as a function of pyruvate concentration, (iv) assume that pyruvate behaves identically to "labile" NOM, and (v) apply this quantitative relationship to natural samples. I find this logic to be somewhat flawed or, at least, justification is incomplete. For example, why should "labile" NOM in the environment, which presumably contains a complex mixture of compounds and function groups, promote HOOH decay with the exact same kinetic rate constant as pyruvate? Does this imply that only "pyruvate-like" compounds (i.e. those with a ketone and/or carboxylic acid function group) are "labile"? In contrast, it has been shown that dissolved lignin can actually decay quite quickly when exposed to uv light (e.g. Spencer et al. 2009 *JGR*). Additionally, it has been shown that highly condensed aromatic and aliphatic organic substrate is rapidly consumed by heterotrophic microbial communities (e.g. Petsch et al. 2001 *Science*; Hemingway et al. 2018 *Science*). However, according to the experimental design of this study, NOM in both of these cases would be considered "recalcitrant."

This makes me wonder what exactly is meant by "labile" and "recalcitrant." Do these terms refer to the bioavailability of NOM and, if so, why were no incubation experiments done to validate that material promoting rapid HOOH decay is actually consumed quickly by heterotrophic communities? (this is certainly true for pyruvate, but what about natural samples?) Or do these terms refer to lability *with respect to reactive oxygen species* and, if so, how would this translate to bioavailability and persistence in the environment?

Additionally, I find the kinetics of these experiments to be poorly described and poorly justified. Most importantly, I am left wondering why the kinetic order of HOOH decay depends on the chemical composition of OM added (i.e. described as zero-order for fluvic acid and lignin but first-order for pyruvate). For example, for the pyruvate case, is HOOH decay first-order with respect to itself, with respect to pyruvate concentration, or both? As this manuscript is written, I *think* the authors treat this as first-order with respect to itself, but this was not tested. Why was an experiment not done in which the NOM concentration was held constant and initial HOOH concentration was varied? This would easily show the reaction order with respect to HOOH concentration (see, for example, Follett et al. 2014 *PNAS* or Hemingway et al. 2017 *Biogeosciences* for mathematical treatment of these results).

If I am interpreting this correctly, then why would HOOH decay be first-order with respect to itself when pyruvate is added but zero-order with respect to itself when lignin or fluvic acid is added? How could this be translated to a natural sample that contains a complex mixture of compounds? Reaction order would need to be known *a priori*. Rather, it seems to me like a more reasonable kinetic model would be one that is zero-order with respect to HOOH concentration and first-order with respect to oxidizable functional groups present in NOM (although the abovementioned test would need to be performed to validate this). If this is true, then HOOH decay could be described as something like:

$$\frac{d[\text{HOOH}]}{dt} = -k_0 - k_1[\text{NOM}] \qquad (1)$$

where $k_0$ is the "intrinsic" zero-order decay rate without NOM present (termed "control" throughout this manuscript) and $k_1$ describes the additional HOOH decay promoted by the presences of NOM and is dependent on NOM chemical composition. This would result in a HOOH half-life that scales inversely with NOM concentration (i.e. $t_{1/2} \propto 1/[\text{NOM}]$). This relationship fits the data reported in Table 1 significantly better

than does the model described in Fig. 2 and Eq. 1-2.

Of course, the model I describe here might not be the best one to describe these data, but as this manuscript is currently written I'm not convinced that it's any worse than the model presented in the text. I recommend the authors provide a sound, theoretical justification for their choice of kinetic model and include a mathematical derivation starting from first principles, in addition to the results of any test experiments needed to verify their choices.

Unfortunately, these issues preclude me from recommending publication of this manuscript without a significant overhaul of the experimental design and data interpretation. I would first recommend that the authors reconsider NOM decay dynamics and move away from the simple idea that some material is "labile" while other material is "recalcitrant." Organic matter decay is now known to be an incredibly complex, dynamic process that depends on heterotroph community composition, interaction with minerals and particles, light, temperature, etc. in addition to chemical composition (see, for example, Schmidt et al. 2011 *Nature* for review). The interpretation taken here – i.e. that decay is solely a function of chemical composition and that compounds can be pooled into labile and recalcitrant fractions – is an outdated one.

This isn't to say that the assay described in this manuscript doesn't have potential – it might. However, hydrogen peroxide decay rates can only speak to the chemical composition of organic matter, which is not the same as lability. I suspect that hydrogen peroxide will decay faster in systems with a higher concentration of carboxylic acids (via formation of peroxyacids) and ketones (via the Baeyer-Villiger oxidation and similar reactions). This seems to be validated by the pyruvate experiments, as pyruvate contains both a carboxylic acid and a ketone. If this is true, then this assay might be useful for describing, in a general sense, the chemical composition of NOM functional groups. However, this would require significantly more validation before being used in environmental samples (e.g. by comparing hydrogen peroxide decay rates with $^{1}$H and $^{13}$C NMR). I believe that this is outside the scope of this manuscript as it is currently

written.

Although I cannot recommend publication of this manuscript without a very serious overhaul, please do not hesitate to contact me for further discussion regarding this review.

Sincerely,

Jordon Hemingway jordon_hemingway@fas.harvard.edu

**References cited in this review**

Follett et al. *PNAS*, **111**, 16706-16711 (2014) (specifically SI text)
Hemingway et al. *Biogeosciences*, **14**, 5099-5114 (2017)
Hemingway et al. *Science*, **360**, 209-212 (2018)
Petsch et al. *Science*, **292**, 1127-1131 (2001)
Schmidt et al. *Nature*, **478**, 49-56 (2011)
Spencer et al. *JGR*, **114**, G03010 (2009)

---

## Author Comment (AC2) · 5 Jun 2018

May 28th, 2018.

To: Biogeosciences – Discussion manuscript - bg-2018-122 Subject: Response to referee 2

We appreciate the valuable time and critical review done by referee 2 and certainly considered the useful comments and suggestions made to improve the manuscript. In this manuscript, our experimental approach was based on previous observations published by Jardim et al (2010), in which it was suggested that H2O2 could be used to distinguish

the difference between organic matter incorporated in waters during flooding periods in Negro River (Amazon Basin), but it was not possible to quantify the amount of LOM. These authors used H2O2 kinetic consumption in two samples (freshwater from Negro River and water fortified with fresh leached soil organic matter). They showed a significant change in the chemical speciation of Hg coordinated by redox conditions in aquatic region studied in the presence of labile organic matter (LOM). In the rainy season, there was a great input of allochthonous natural organic matter (NOM) in aquatic bodies, and this NOM, considered fresh and reactive, would be able to scavenge H2O2 naturally photogenerated in the water column, influencing directly the oxidation conditions in this environment. Thus, this comprises one of the direct effects caused by the presence of LOM. Please, find ahead our answers for specific questions given in point by point below.

Referee: This makes me wonder what exactly is meant by "labile" and "recalcitrant." Do these terms refer to the bioavailability of NOM and, if so, why were no incubation experiments done to validate that material promoting rapid HOOH decay is actually consumed quickly by heterotrophic communities? (this is certainly true for pyruvate, but what about natural samples?) Or do these terms refer to lability with respect to reactive oxygen species and, if so, how would this translate to bioavailability and persistence in the environment?

Authors: We aimed at the possibility of quantifying labile and recalcitrant organic matter in freshwater samples. This objective was based on the importance that NOM plays in aquatic environment. It is known that NOM plays a relevant role in photoreactions, forming reactive species, or even scavenging these species. It is also primary source of biota and it is able to complex or adsorb other species as well. In this context, we would like to try measured the different degrees of reactivity of NOM according the characteristics and considering our definition. We denominated LOM as NOM that was few oxidized or degraded and it is still able to react as a scavenger of oxidant species in aquatic systems. On the other hand, recalcitrant organic matter (ROM) is the fraction that had already suffered oxidation, and it is less reactive towards oxidant species, such as H2O2. Our approach is different from classical methods used to distinguish organic matter degraded by microorganism or chemically, such as the ones used in the biochemical oxygen demand (BOD) and chemical oxygen demand (COD) measurements, respectively. We agree with the referee that the way of LOM and ROM were defined in this work, it could be reformulated in the new version of the manuscript. In our attempt, we led the lability and recalcitrance concepts through the chemical approach, trying to reach a simpler approach than the protocols currently used to determine labile fraction, that consider it as biodegradable fraction of NOM, hence they always include bioassays. Considering that the concept used by us to describe the lability and the points highlighted by the referee, we agree that our experimental approach was not enough to distinguish the labile fraction of NOM, but rather those part of NOM that is able to scavenger the H2O2, which might to be construed as the NOM components that are resemblance to our model (i.e. pyruvate), in terms of the reactive groups present, as suggested by the referee as well. Therefore, certainly we will improve the discussion over this subject using this experimental approach and the other amendment.

Referee: Additionally, I find the kinetics of these experiments to be poorly described and poorly justified. Most importantly, I am left wondering why the kinetic order of HOOH decay depends on the chemical composition of OM added (i.e. described as zero-order for fluvic acid and lignin but first-order for pyruvate). For example, for the pyruvate case, is HOOH decay first-order with respect to itself, with respect to pyruvate concentration, or both? As this manuscript is written, I think the authors treat this as first-order with respect to itself, but this was not tested. Why was an experiment not done in which the NOM concentration was held constant and initial HOOH concentration was varied?

Authors: On this questioning about the kinetic of decay of H2O2, we agree that is feasible to realize an experiment using a range o H2O2 and exceed of pyruvate to

confirm the kinetic order.

Referee: I recommend the authors provide a sound, theoretical justification for their choice of kinetic model and include a mathematical derivation starting from first principles, in addition to the results of any test experiments needed to verify their choices. [. . .] If I am interpreting this correctly, then why would HOOH decay be first-order with respect to itself when pyruvate is added but zero-order with respect to itself when lignin or fluvic acid is added? How could this be translated to a natural sample that contains a complex mixture of compounds? Reaction order would need to be known a priori. Rather, it seems to me like a more reasonable kinetic model would be one that is zero order with respect to HOOH concentration and first-order with respect to oxidizable functional groups present in NOM (although the abovementioned test would need to be performed to validate this). If this is true, then HOOH decay could be described as something like: *equation 1 (see comments of referee 2) where k0 is the "intrinsic" zero-order decay rate without NOM present (termed "control" throughout this manuscript) and k1 describes the additional HOOH decay promoted by the presences of NOM and is dependent on NOM chemical composition. This would result in a HOOH half-life that scales inversely with NOM concentration (i.e. t1=2 / 1=[NOM]). This relationship fits the data reported in Table 1 significantly better than does the model described in Fig. 2 and Eq. 1-2. Of course, the model I describe here might not be the best one to describe these data, but as this manuscript is currently written I'm not convinced that it's any worse than the model presented in the text. I recommend the authors provide a sound, theoretical justification for their choice of kinetic model and include a mathematical derivation starting from first principles, in addition to the results of any test experiments needed to verify their choices.

Authors: It is important to add that kinetic models employed here to determine the order and consequently half-lives of $H2O2$ were based on the mathematical strict sense of the classical kinetic laws, as an exponential decay formula, as to fit the data, and it was not our attempt with this experimental approach discuss about any specific chemical

mechanism behind these reactions, once our focus is to apply the mathematical formalism, only in its empirical form, to general and more complex systems such as natural aquatic samples. We have chosen the order of the exponential function, for each case, using the correlation of the fit as the mandatory parameter, so we could decide the best order for each model compound used. Empirically, we found that in presence of pyruvate, peroxide reacts in a first order kinetic decay. In truth, in those cases, we have an excess of pyruvate allowing a treatment as pseudo-first order kinetics. We also have found that in the absence of pyruvate the systems decay in a zero order fashion which, as we have observed, is characteristic of the HOOH spontaneous decay in the system condition, and it is quite slow compared to the reactions in presence of pyruvate, so for that reason it should be neglected from the equation. Our first conclusion was that all those models used (lignin, fulvic acid, and others not mentioned in this work), except pyruvate, cannot be used as a LOM model, neither as recalcitrant model, once what we see in the zero order behavior, probably is the spontaneous decomposition of HOOH in water. Once we have realized that, we start to look at the reason why pyruvate could behave as a model for LOM. From that we came with follow interpretation based in a solely chemical selection approach:

Figure 1.

In the basis of the mechanism shown in Figure 1 we have made ab initio calculations for roughly (Hartree-Fock level – HF/6-31++G(d,p) – of theory using C-PCM as solvation model to simulate aqueous system) estimate the Gibbs Free Energy of Activation for the reaction (G‡) and we have found a value of ∼24 kcal mol-1 (at 25 °C) for this main barrier (rate determining step).

Figure 2.

Reflecting on the basis of the referee's questions, in our hypothesis, any specie which could react with HOOH with an activation barrier lower or equal to this value (24 kcal mol-1) should be considered LOM, and that should be reached only for "easily" organic

oxidable matter. That should install a chemical definition for LOM, it means that once we have that amount of energy, a threshold stipulated on the basis of the pyruvate reaction, any specie that react faster or in equal rate should be considered LOM. For this reason, pyruvate has a behavior that we could correlate to the LOM in the environmental samples analyzed. It follows the Cartesian coordinates of the transition state, pyruvate and HOO- used in the calculation CPCM/HF/6-31++G(d,p) performed in GAMESS-US version 1 May 2013 (R1).

Transition State C 6.0 1.6816350116 0.1010684975 -0.2277477680 O 8.0 2.1332403247 -0.4627711238 0.7839124009 O 8.0 2.3445793973 0.7031280298 -1.0817602132 C 6.0 0.1467425077 0.0145001255 -0.4409410642 O 8.0 -0.3233234672 0.2048176881 -1.5749461202 C 6.0 -0.6180775921 -0.8880991464 0.5001816824 O 8.0 -0.1290602750 1.6670036824 0.5265217665 O 8.0 -1.1608590282 2.3091250331 -0.2028123400 H 1.0 -1.6690724085 -0.6319647769 0.4508054964 H 1.0 -0.2642339719 -0.8189532239 1.5155537825 H 1.0 -0.4969221131 -1.9142434582 0.1576182172 H 1.0 -1.1713883854 1.8101786728 -1.0157058403

Pyruvate C 6.0 1.6935563717 0.0186054902 -0.2705855508 O 8.0 2.0945133541 -0.0529859412 0.8957945494 O 8.0 2.3570246488 0.1042454662 -1.3071337138 C 6.0 0.1555022243 0.0193979230 -0.4143599551 O 8.0 -0.3833560760 0.8081793314 -1.1419747000 C 6.0 -0.6146609989 -0.9645362969 0.4225977542 H 1.0 -1.6730939503 -0.8819106367 0.2158895184 H 1.0 -0.4215626761 -0.7698612741 1.4705418768 H 1.0 -0.2733428976 -1.9736340619 0.2119002211

HOO- O 8.0 -0.1123690206 1.6640975373 0.5502444474 O 8.0 -1.1627181376 2.3027578858 -0.2161034494 H 1.0 -1.1862228418 1.8194445769 -1.0261409980

Thank you. Sincerely yours, Isabela C. Constantino and co-workers.
* * *
[Figure]

**Fig. 1.** Suggested mechanism for the decarboxylation of pyruvate from HOOH. Inspired by Phys.Chem.Chem.Phys.,2017, 19, 19316 and Lopalco et al., J. Pharm. Sci..2015.DOI 10.1002/jps.24653

[Figure]

1.93

**Fig. 2.** Transition State for the determining step of the decarboxylation of pyruvate by HOOH, found using HF/6-31+G(d,p) and C-PCM for simulating the aqueous environment. 666.

[Figure]